# NusG is an intrinsic transcription termination factor that stimulates motility and coordinates gene expression with NusA

Zachary F Mandell[1], Reid T Oshiro[2], Alexander V Yakhnin[3], Rishi Vishwakarma[1], Mikhail Kashlev[3], Daniel B Kearns[2], Paul Babitzke[1]*

[1]Department of Biochemistry and Molecular Biology, Center for RNA Molecular Biology, Pennsylvania State University, University Park, United States; [2]Department of Biology, Indiana University, Bloomington, United States; [3]NCI RNA Biology Laboratory, Center for Cancer Research, NCI, Frederick, United States

**Abstract** NusA and NusG are transcription factors that stimulate RNA polymerase pausing in *Bacillus subtilis*. While NusA was known to function as an intrinsic termination factor in *B. subtilis*, the role of NusG in this process was unknown. To examine the individual and combinatorial roles that NusA and NusG play in intrinsic termination, Term-seq was conducted in wild type, NusA depletion, *ΔnusG*, and NusA depletion *ΔnusG* strains. We determined that NusG functions as an intrinsic termination factor that works alone and cooperatively with NusA to facilitate termination at 88% of the 1400 identified intrinsic terminators. Our results indicate that NusG stimulates a sequence-specific pause that assists in the completion of suboptimal terminator hairpins with weak terminal A-U and G-U base pairs at the bottom of the stem. Loss of NusA and NusG leads to global misregulation of gene expression and loss of NusG results in flagella and swimming motility defects.

*For correspondence:
pxb28@psu.edu

**Competing interests:** The authors declare that no competing interests exist.

## Introduction

The transcription cycle can be subdivided into initiation, elongation, and termination. Regulation of initiation by a variety of DNA binding proteins is well established, while elongation is regulated by auxiliary transcription factors that interact with RNA polymerase (RNAP), including the elongation factors NusA and NusG (*Mondal et al., 2017*). Transcription termination demarcates the 3' ends of transcription units, and misregulation of this process can result in spurious sense and/or antisense transcription (*Roberts, 2019*; *Mondal et al., 2016*). Termination in bacterial systems is known to proceed via two distinct mechanisms. One mechanism is Rho-dependent termination, which requires the activity of Rho, a hexameric ATP-dependent RNA translocase (*Roberts, 2019*). The other mechanism is intrinsic termination, which is generally assumed to not require the activity of additional protein factors and as such is also frequently referred to as factor-independent termination (*Roberts, 2019*). An intrinsic terminator is composed of a GC-rich hairpin followed immediately by a U-rich tract, both of which define the point of termination (POT) (*Roberts, 2019*). Completion of the terminator hairpin can induce transcript release via hybrid shearing and/or hyper-translocation depending on the transcriptomic context (*Komissarova et al., 2002*; *Roberts, 2019*).

NusA is a conserved bacterial transcription elongation factor that is essential for cellular viability in both *Bacillus subtilis* and *Escherichia coli*. This protein factor binds to the flap-tip domain of the β subunit of RNAP via its N-terminal domain (NTD) (*Guo et al., 2018*). Once bound, NusA can directly interact with RNA using its S1, KH1, and KH2 domains (*Guo et al., 2018*). In *E. coli*, the binding of

these domains to RNA elements allows NusA in combination with several other Nus proteins to serve as an antitermination factor when transcribing rRNA or bacteriophage λ sequences (*Nodwell and Greenblatt, 1991*; *Vogel and Jensen, 1997*). Also, the NTD of NusA can provide an additional set of positively charged residues outside of the RNA exit channel, extending this cavity while stabilizing the nucleation of RNA hairpins (*Guo et al., 2018*). Moreover, the binding of NusA results in an allosteric widening of the RNA exit channel of RNAP, which can then more readily accommodate RNA duplexes (*Guo et al., 2018*). The ability of NusA to promote the nucleation and formation of hairpins within the RNA exit channel allows this factor to serve as both a pausing factor and a termination factor (*Guo et al., 2018*). Interestingly, the antitermination activity of RNAP during transcription of rRNA operons involves suppression of NusA-stimulated pausing (*Huang et al., 2020*).

NusG, or SPT5 in archaeal and eukaryotic organisms, is the only universally conserved transcription factor. Bacterial NusG has two domains: the N-terminal NGN domain, which binds to the clamp helices of the β′ subunit of RNAP, and the KOW domain, which is connected to the NGN domain via a flexible linker and is free to interact with various regulatory partners (*Liu and Steitz, 2017*). In *E. coli*, NusG is an anti-pausing factor and a core regulator of transcriptional polarity via the ability of its KOW domain to interact with either Rho or ribosomal protein S10 (*Mooney et al., 2009*; *Tomar and Artsimovitch, 2013*). Also, the KOW domain of *E. coli* NusG can interact with the S1 domain of NusA when coordinated by NusE and λN during bacteriophage λ antitermination (*Krupp et al., 2019*). In contrast, *B. subtilis* NusG is a sequence-specific pausing factor due to the ability of its NGN domain to make direct contacts with the non-template DNA (ntDNA) strand within the transcription bubble (*Yakhnin et al., 2016*). This interaction results in a pause when the NGN domain encounters a stretch of T residues at critically conserved positions, as observed in 1600 NusG-dependent pause sites genome-wide (*Yakhnin et al., 2020*). T residues in the ntDNA strand correspond to U residues in the nascent transcript. Although pausing is thought to be a fundamental prerequisite to termination, the role of NusG in intrinsic termination has not been investigated in *B. subtilis*.

While intrinsic termination does not require additional auxiliary protein factors, it has been known for many years that NusA can stimulate intrinsic termination of *E. coli* and *B. subtilis* RNAP in vitro (*Greenblatt et al., 1981*; *Schmidt and Chamberlin, 1987*; *Bermúdez-Cruz et al., 1999*; *Yakhnin and Babitzke, 2002*). The ability of mycobacterial NusG to stimulate intrinsic termination in vitro has also been reported (*Czyz et al., 2014*). More recently it was determined that NusA stimulates intrinsic termination in vivo on a global level in *B. subtilis*, with 232 intrinsic terminators classified as NusA-dependent (*Mondal et al., 2016*). In the current study we show that *B. subtilis* NusG also functions as an intrinsic termination factor in vivo, and that NusA and NusG cooperatively stimulate intrinsic termination on an unexpectedly large scale, with only 12% of all identified intrinsic terminators continuing to terminate efficiently in the absence of these two proteins. Our results suggest a model in which NusG-dependent pausing plays a vital role in NusG-dependent termination, and that the absence of NusG results in the misregulation of global gene expression and altered cellular physiology and behavior.

## Results

### NusG and NusA cooperatively stimulate intrinsic termination in vivo

For this study, we used a $nusA_{dep}$ strain in which NusA was solely generated exogenously from an IPTG-inducible promoter (*Mondal et al., 2016*). Thus, growth in the presence of IPTG results in wild-type (WT) levels of NusA, whereas growth in the absence of IPTG results in depletion of NusA to less than 2% of WT levels within four cell generations as shown via Western blot (*Figure 1—figure supplement 1*). By performing our studies with $nusA_{dep}$ and $nusA_{dep}\Delta nusG$ *B. subtilis* strains ± IPTG, we were able to mimic WT ($nusA_{dep}$, +IPTG), NusA depletion ($nusA_{dep}$, –IPTG), *nusG* deletion ($nusA_{dep} \Delta nusG$, +IPTG), and NusA depletion *nusG* deletion ($nusA_{dep} \Delta nusG$, –IPTG) conditions. To simplify the discussion, we will refer to these four conditions as WT, $nusA_{dep}$, $\Delta nusG$, and $nusA_{dep} \Delta nusG$ strains.

Term-seq is a bulk functional genomics assay that allows for the identification of all 3′ ends within a transcriptome via the ligation of a unique RNA oligonucleotide to the 3′ end of all transcripts

isolated from a bacterial culture (*Mondal et al., 2016*; *Dar et al., 2016*). This ligation effectively preserves the authentic 3' ends of all ligated transcripts, allowing for the computational identification of all authentic 3' ends after sequencing (*Mondal et al., 2016*). To study the impact of NusA and NusG on intrinsic termination, we conducted Term-seq in duplicate in the WT, $nusA_{dep}$, $\Delta nusG$, and $nusA_{dep}$ $\Delta nusG$ strains. For each genomic region found to contain a transcript 3' terminus, there were often multiple adjacent 3' ends (*Mondal et al., 2016*). For our purposes, only the most abundant 3' end within each region was included in the subsequent terminator analysis. Each 3' end containing the core intrinsic terminator modules (RNA hairpin and U-rich tract) in the upstream sequence were categorized as potential intrinsic terminators, and a potential intrinsic terminator was confirmed to terminate in vivo only in cases where the termination efficiency (%T) at this nucleotide (nt) was ≥5 in the WT strain (see 'Materials and methods'). Using this system, we identified 4657 3' ends in the WT strain (*Supplementary file 1*), 1400 of which were categorized as intrinsic terminators (*Supplementary file 2*). To benchmark the results of our assay, we compared the locations of all intrinsic terminators identified in this study, to all intrinsic terminators identified previously by Term-seq in WT *B. subtilis* grown in Minimal-ACH media (*Mondal et al., 2016*), and all intrinsic terminators identified by the in silico intrinsic terminator prediction tool TransTermHP applied to the *B. subtilis* genome (*Figure 1—figure supplement 2*; *Kingsford et al., 2007*). This analysis showed a high level of overlap between these datasets, with 937 terminators being conserved in all three datasets and 1329 terminators being shared between our dataset and at least one other dataset. We also found 1123 intrinsic terminators shared between this study and a previous intrinsic terminator study conducted via Rend-Seq, a different 3' end mapping strategy (*Lalanne et al., 2018*).

The %T was calculated for each of the 1400 intrinsic terminators in each strain. Violin plots overlayed with box plots were constructed to view the distribution of these data, and the data collected from each strain was compared via Wilcoxon signed-rank testing (*Figure 1A*, *Supplementary file 2*, *Supplementary file 3*). NusG stimulated intrinsic termination to a similar extent as NusA, with an ~ 22% drop in median %T in the $nusA_{dep}$ and $\Delta nusG$ strains when compared to the WT strain. Loss of both NusA and NusG in the $nusA_{dep}$ $\Delta nusG$ strain resulted in a drastic termination defect, with the median %T falling 55%. Change in %T upon the loss of NusA and/or NusG (Δ%T) was calculated for all intrinsic terminators in each mutant strain (*Supplementary file 2*). Based on our previously established categorization scheme, an intrinsic terminator was categorized as 'dependent' on NusA and/or NusG when the Δ%T ≥ 25, and 'independent' of NusA and/or NusG when 10 ≥ Δ%T ≥ −10 (*Mondal et al., 2016*). Remarkably, only 12% of all intrinsic terminators were categorized as independent in the $nusA_{dep}$ $\Delta nusG$ strain. To further assess the scope of the relationship between NusA and NusG on intrinsic termination, the overlap of intrinsic terminators categorized as dependent in each strain was organized into a Venn diagram and various intrinsic terminator subpopulations were identified (*Figure 1B*). Terminators that were classified as dependent in only the $nusA_{dep}$ single mutant and $nusA_{dep}$ $\Delta nusG$ double mutant strains were categorized as requiring NusA (Req A), while terminators that were classified as dependent in only the $\Delta nusG$ single mutant and $nusA_{dep}$ $\Delta nusG$ double mutant strains were categorized as requiring NusG (Req G). Terminators that were categorized as dependent in all three strains were classified as requiring both NusA and NusG (Req A and G). A large number of terminators were only categorized as dependent in the double mutant strain, indicating they were able to terminate efficiently when either NusA or NusG was present in the cell, but not when both were absent. As such, this subpopulation was categorized as requiring either NusA or NusG (Req A or G). Intriguingly, 65% of all intrinsic terminators depicted in *Figure 1B* are present in the Req A and G or in the Req A or G subpopulations, clearly illustrating a large functional overlap between NusA and NusG on intrinsic termination.

The predicted hairpin stability for all intrinsic terminators within each identified subpopulation, including the subpopulation found to terminate strongly in the WT strain (%T ≥ 70) and be independent of both protein factors (strong and independent [SI]), was calculated and this data was organized into violin plots overlayed with box plots (*Figure 1C*). The data from each subpopulation was then compared via Mann-Whitney U testing (*Supplementary file 3*). In parallel, sequence logos were generated from the 9 nt regions immediately downstream of the predicted hairpin for terminators within each subpopulation (i.e., the predicted U-rich tract) and each sequence logo was compared using DiffLogo, a tool that computes and displays the per-nucleotide Jensen-Shannon divergence for a set of sequence motifs in a pairwise fashion (*Figure 1D*, *Figure 1—figure supplement 3*; *Nettling et al., 2015*). Our terminator prediction system considered the hairpin to end

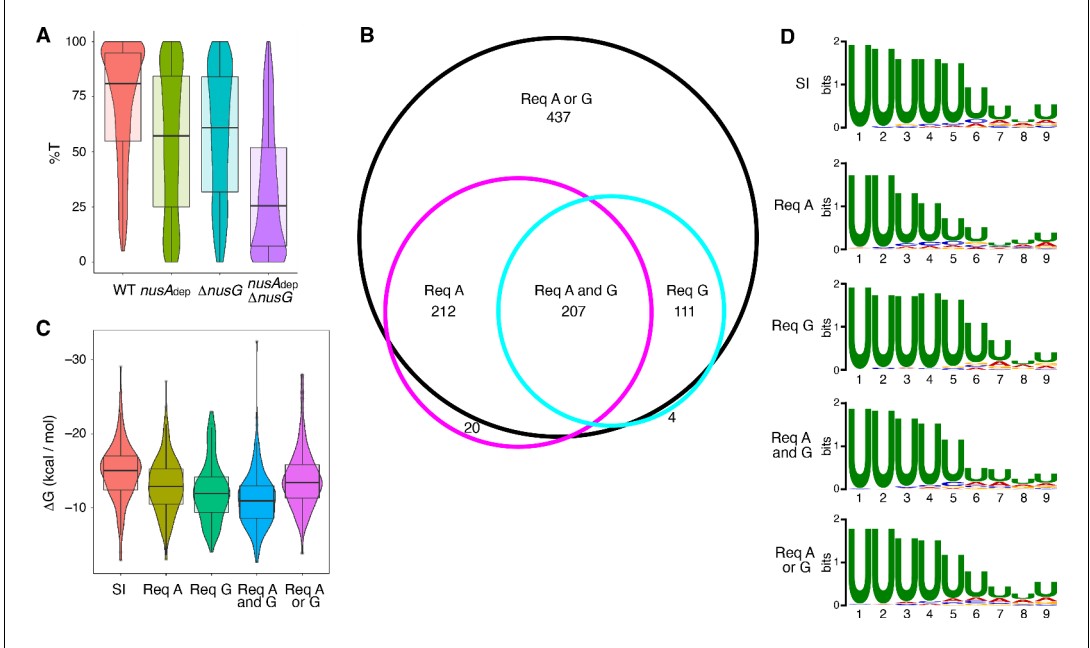

**Figure 1.** NusG is an intrinsic termination factor that works with NusA to stimulate suboptimal terminators. (A) Violin plots overlaid with box plots showing the distribution of termination efficiency (%T) in wild-type (WT), *nusA*$_{dep}$, Δ*nusG*, and *nusA*$_{dep}$ Δ*nusG* strains. For this and all box plots, boundaries of the box designate the interquartile range (IQR), while upper/lower whiskers extend from the 75th/25th percentile to the largest/smallest value no further than 1.5*IQR in either direction. All pairwise p-values can be obtained from *Supplementary file 3*. (B) Venn diagram showing the number and overlap of terminators that were classified as being dependent (Δ%T ≥ 25) on NusA (Req A) and/or NusG (Req G). The black circle contains all terminators that were classified as dependent in the *nusA*$_{dep}$ Δ*nusG* strain, the magenta circle contains all terminators that were classified as dependent in the *nusA*$_{dep}$ strain, and the cyan circle contains all terminators that were classified as dependent in the Δ*nusG* strain. Intrinsic terminator subpopulations that require NusA and/or NusG in any fashion to terminate efficiently are specified. (C) Violin plots overlaid with box plots showing the distribution of predicted hairpin strength as reported in ΔG (kcal/mol) for all identified subpopulations including the strong and independent (SI) terminators. All pairwise p-values can be obtained from *Supplementary file 3*. (D) Sequence logos of the U-rich tracts generated from the nine nucleotide (nt) window downstream of the predicted hairpins for all identified subpopulations including the SI terminators. All pairwise comparisons can be found in *Supplementary file 3*.

The online version of this article includes the following figure supplement(s) for figure 1:

**Figure supplement 1.** Western blot analysis of NusA depletion.
**Figure supplement 2.** Benchmarking the intrinsic terminators identified in this study.
**Figure supplement 3.** Pairwise comparative analysis of U-rich tract sequence logos.
**Figure supplement 4.** Intrinsic terminator hairpin stem length and loop length.
**Figure supplement 5.** Transcriptomics data showing that Term-seq replicates are highly correlated and data from each strain is distinct.

before the first U residue and any contribution of A-U base pairing to terminator hairpin stability was not considered. It was found that terminators that require NusG in any fashion have weaker predicted hairpins than SI terminators, akin to Req A terminators, while terminators that require both NusA and NusG have the weakest hairpins (*Figure 1C*). In addition, terminators that require NusG in any fashion exhibit a stronger enrichment of U residues downstream of the predicted hairpin than Req A terminators (*Figure 1D*, *Figure 1—figure supplement 3*). This strong U enrichment is in line with the hypothesis that NusG stimulates intrinsic termination through its role in pausing (see below). Analyzing the distribution of predicted terminator hairpin stem lengths from each subpopulation with Fisher-Pitman permutation testing shows that SI terminators have longer hairpin stems than all subpopulations except Req A or G terminators, with a median of 10 nt (*Figure 1—figure supplement 4*, *Supplementary file 3*). This difference likely contributes to the observation that SI terminators have the strongest hairpins (*Figure 1C*, *Supplementary file 3*). A similar analysis examining terminator hairpin loop length via asymptotic K-sample Fisher-Pitman permutation testing showed no appreciable difference between these subpopulations (*Figure 1—figure supplement 4*).

### The NGN domain of NusG promotes pausing at the POT

Our in vivo results indicated that NusG stimulates intrinsic termination cooperatively with NusA. To examine this phenomenon further while exploring a potential mechanism, we cloned six terminators found to require NusA and/or NusG in vivo for in vitro experimentation. We first examined the *yetJ* terminator (Req A and G) (*Figure 2A and B*) and the *ktrD* terminator (Req G) (*Figure 2D and E*). To maintain a logical consistency between our in vivo and in vitro data, each in vivo condition is labeled to indicate the elongation factors that were present in the cell. Single-round termination assays were conducted with these two terminators ± NusA and/or ± NusG. Experiments were also performed with the WT NusG NGN domain, because this domain was found to be sufficient to recapitulate all features of NusG-dependent pausing (*Yakhnin et al., 2016*). In addition, a mutant NGN domain in which the residues responsible for eliciting a pause were replaced with the corresponding *E. coli* residues (Y77H/N81S/T82V) was included in the analysis. This mutant NGN domain was unable to

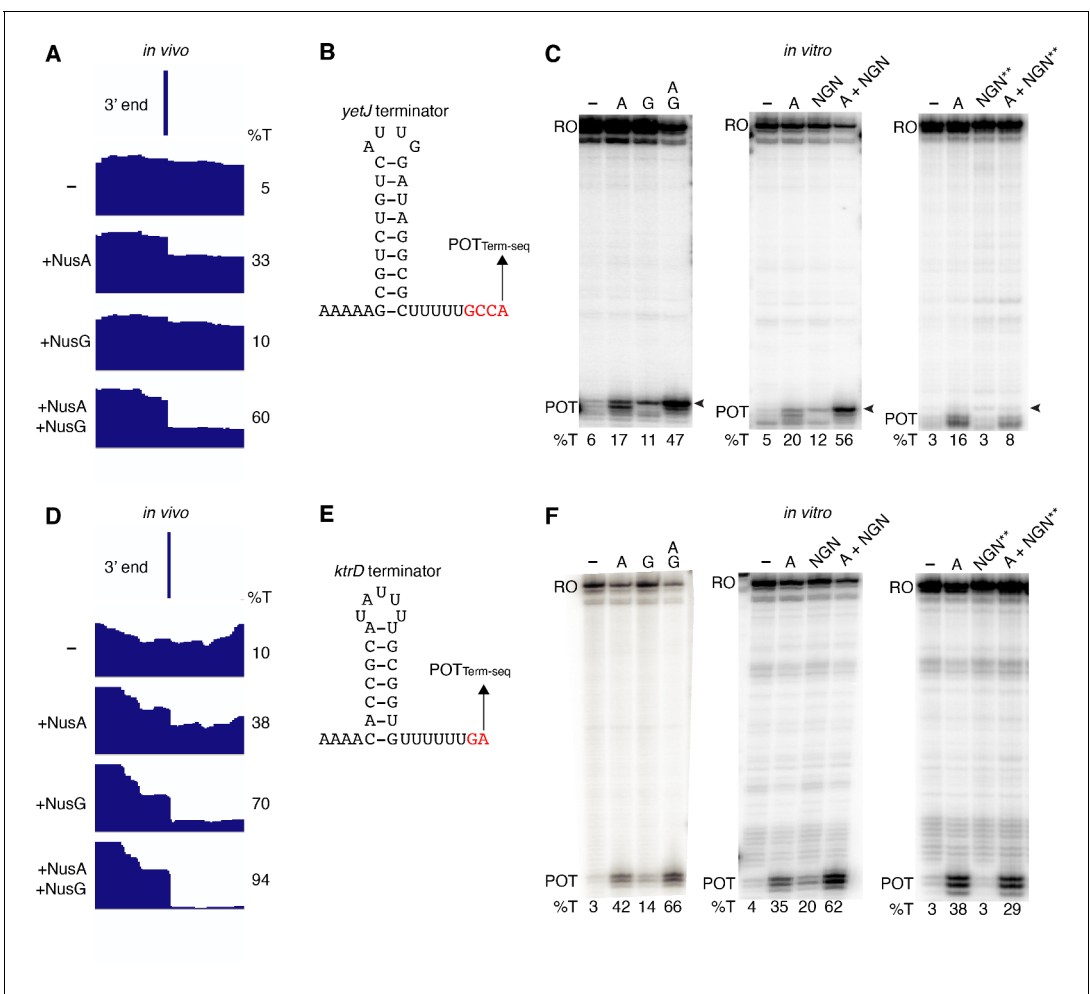

**Figure 2.** NusG stimulates intrinsic termination via its NGN domain. (**A**) IGV screenshot of a genomic window centered around the *yetJ* terminator. Top track is the 3' end identified by Term-seq. Bottom tracks are the RNA-seq coverage data for the $nusA_{dep} \Delta nusG$ (-), $\Delta nusG$ (+NusA), $nusA_{dep}$ (+NusG), and wild-type (WT) (+NusA +NusG) strains. %T in each strain is shown on the right of each track. Transcription proceeds from left to right. (**B**) *YetJ* terminator showing the point of termination identified by Term-seq in vivo ($POT_{Term-seq}$). Disruptions in the U-rich tract are shown in red. The upstream A tract is also shown. (**C**) Single-round in vitro termination assay with the *yetJ* terminator. Experiments were performed in the absence (–) or presence of NusA (**A**), NusG (**G**), WT NGN domain, and/or mutant NGN domain as indicated (mutant NGN domain is signified as NGN**). Positions of terminated (POT) and run-off (RO) transcripts are marked. The arrowhead marks the most distal POT. %T is shown below each lane. (**D–F**) Identical to panels (**A–C**) except that it is the *ktrD* terminator.

The online version of this article includes the following figure supplement(s) for figure 2:

**Figure supplement 1.** Template design for in vitro transcription.

promote pausing at the NusG-dependent TTNTTT pause motif found in the *trp* leader (*Yakhnin et al., 2016*). NusG recapitulated the stimulatory effect on termination of the *yetJ* and *ktrD* terminators, as well as the cooperative effect between NusA and NusG that we observed in vivo (*Figure 2C and F*). Moreover, the highly similar termination patterns observed when using either full-length NusG or the NGN domain (± NusA) demonstrate that this phenomenon can be fully attributed to the NGN domain of NusG (*Figure 2C and F*). The lack of either a stimulatory effect on termination or a cooperative effect on termination with NusA by the mutant NGN domain indicates that the effect of NusG on termination can be explained through its role as a pausing factor (*Figure 2C and F*).

Our results with the NGN domain showed that NusG exerts its effect on intrinsic termination through its role in pausing. To further explore the connection between NusG-dependent pausing and termination, single-round in vitro transcription time course (pausing) assays were conducted with the *ktrD* terminator ± NusG (*Figure 3A and B*). Results from this assay revealed three consecutive NusG-dependent transcription products that were 9, 10, and 11 nt downstream from the predicted terminator hairpin. Comparing the RNA species in the time course lanes with the 30 min termination lane shows that the product 9 nt from the predicted hairpin is a NusG-dependent termination site (POT$_1$), the product 10 nt from the predicted hairpin is a NusG-dependent pause site and a NusG-dependent termination site (Pause$_1$/POT$_2$), and the product 11 nt from the predicted hairpin is a NusG-dependent pause site (Pause$_2$). Quantifying the relative intensity of Pause$_1$/POT$_2$ at the 30

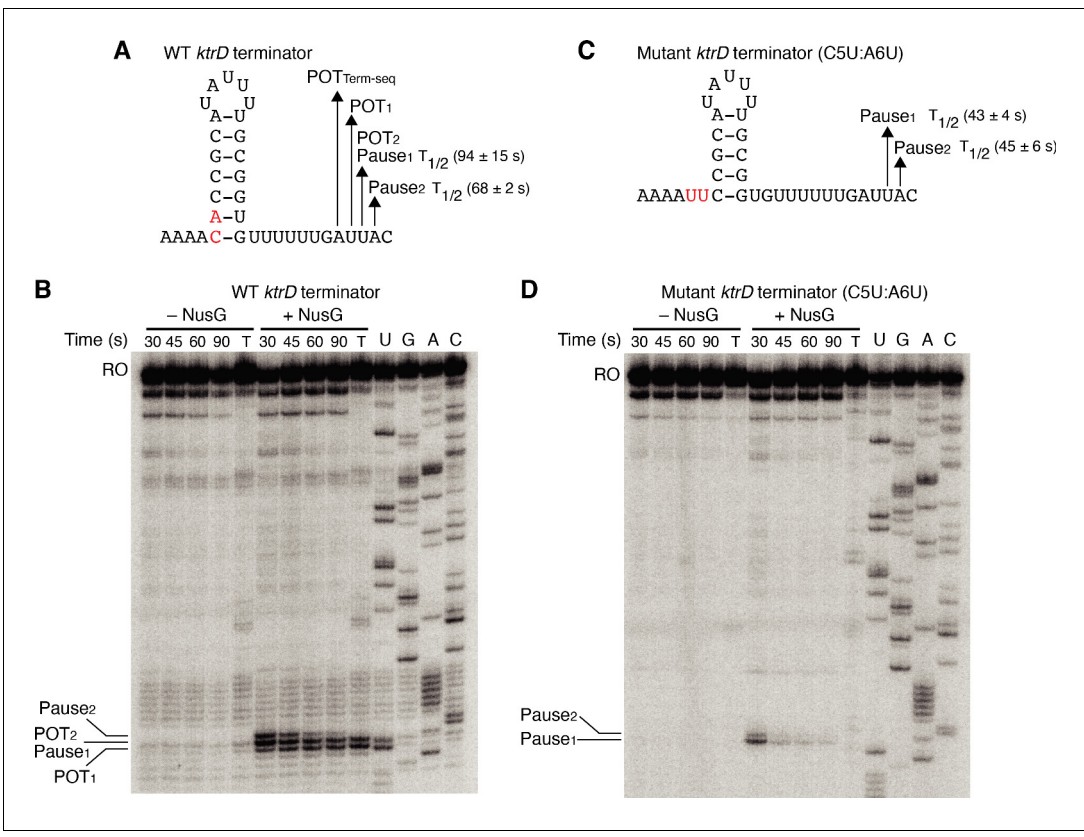

**Figure 3.** NusG stimulates termination through its role as a sequence-specific pause factor. (**A**) *KtrD* terminator showing the points of termination (POT$_1$ and POT$_2$) and pause sites (Pause$_1$ and Pause$_2$) identified in vitro. The POT identified in vivo by Term-seq is also specified (POT$_{Term-seq}$). The upstream A tract is also shown. The average half-life (T$_{1/2}$) of each pause ± standard deviation based on in vitro pausing data are specified to the right of each pause in parentheses. (**B**) Single-round in vitro pausing and termination assay using the wild-type (WT) *ktrD* terminator ± NusG. Time points of elongation are indicated above each lane. T, 30 min termination assay. RNA sequencing lanes (U, G, A, C) are labeled. Positions of NusG-dependent pause bands, termination sites, and run-off (RO) transcripts are marked. (**C, D**) Identical to (**A–B**) except that it is the C5U:A6U mutant *ktrD* terminator. The WT (**A**) and mutated (**C**) residues are highlighted in red.

s vs. 90 s time points showed that the intensity of the band at 30 s was ~1.5-fold higher than the intensity at 90 s, suggesting that a fraction of this RNA species chased to longer transcripts (i.e., a pause), while another fraction did not chase (i.e., terminated).

The hairpin to 3′ end distance for intrinsic terminators is 7–9 nt, whereas this distance is 11–12 nt for pause sites (*Mondal et al., 2016*; *Yakhnin et al., 2020*; *Ray-Soni et al., 2016*). Our results with the *ktrD* terminator indicate that NusG can elicit a pause at two consecutive positions, but only one of these positions contains the requisite elements to induce transcript release. This distinction likely depends on the hairpin to 3′ end distance achieved at each position before readthrough. To test this possibility, we mutated the *ktrD* terminator by substituting C5 and A6 with U residues to reduce the length of the hairpin and to increase the hairpin to 3′ end distance by 2 nt while leaving the NusG-dependent pause motif intact (*Figure 3C*). These mutations altered the transcription profile of this template such that pausing still occurred at positions identical to the positions of Pause$_1$ and Pause$_2$ on the WT template (*Figure 3D*). These results indicate that the NusG-dependent pause motif upstream of the 3′ end is sufficient to elicit a pause and that the position of this pause is set by the motif. Calculating the half-life of both pauses on each template revealed that the duration of the pauses on the mutant template was shorter than the pauses on the WT template, implying that a NusG-dependent pause can be stabilized to different degrees by hairpins of different lengths and strengths. Interestingly, a fraction of RNAP at Pause$_1$ on the mutant template remained paused until at least 90 s, suggesting that NusG-dependent pausing at this position is long-lived. Termination no longer occurred on the mutant template, demonstrating that we successfully converted a NusG-dependent terminator into a NusG-dependent pause site by extending the hairpin to 3′ end distance by 2 nt.

## NusG stimulates pausing at terminators with weak base pairs at the bottom of the hairpin stem and distal U-rich tract interruptions

Termination occurring 7–9 nt downstream of the hairpin is a biophysical constraint set by the length of the RNA-DNA hybrid when RNAP is in the post-translocated state (*Ray-Soni et al., 2016*). The presence of terminated RNA species 10 nt downstream of the predicted *ktrD* terminator hairpin implies that this hairpin in reality extends further via A-U base pairing. Terminators with hairpin stems that contain multiple consecutive terminal A-U base pairs are highly atypical, yet a similar phenomenon was observed for the *yetJ* (Req A and G), *fur* (Req A and G), *yneF* (Req A and G), and *yxiS* (Req G) terminators (*Figure 4*, *Figure 4—figure supplement 1*). Notably, in all cases NusG effectively stimulated termination in vitro at a POT that utilized hairpins with two to four consecutive A-U base pairs at the bottom of the hairpin stem. Interestingly, we also found that NusG stimulated the *fur* terminator in vitro to terminate at a position 9 nt downstream of a hairpin that utilized a G-U base pair at the bottom of the hairpin stem (*Figure 4D*).

While the sequence logos generated from our in vivo terminator hairpin predictions shows an enrichment of U residues immediately downstream of the predicted hairpin for terminators that depend on NusG (*Figure 1D*), our in vitro 3′ end mapping suggests that the upstream portion of this U-rich tract is actually present at the base of the terminator hairpin. Comparison of the NusG-dependent pause motif with the terminator sequences confirmed in vitro shows that each terminator contains U residues at positions that correspond to the T residues in the ntDNA strand that are most critical for NusG-dependent pausing (*Figure 5*, green residues) (*Yakhnin et al., 2020*). Moreover, the revised U-rich tract sequences show that NusG-dependent terminators frequently contain distal U-rich tract interruptions, akin to NusA-dependent terminators (*Figure 5*, red residues) (*Mondal et al., 2016*).

We found that termination in vitro occurred at a position 1–3 nt further downstream than the 3′ end identified in vivo by Term-seq on all templates tested (*Figures 3B*, *4B and E*, *Figure 4—figure supplement 1*). We hypothesized that the discrepancy between the 3′ ends identified in vivo and those identified in vitro could be attributed to trimming of these terminated transcripts by a 3′–5′ exoribonuclease. If this were true, there should be many instances of steady-state 3′ ends that mapped upstream of the corresponding 3′ end at the authentic point of transcript release. To determine whether this actually happens to an appreciable extent in vivo, we turned to RNET-seq, a bulk functional genomics assay that can be used to map the genomic position of all actively transcribing RNAPs (*Yakhnin et al., 2020*). Importantly, the nascent 3′ end information collected by RNET-seq is not affected by post-transcriptional RNA processing, while the 3′ ends identified by Term-seq have

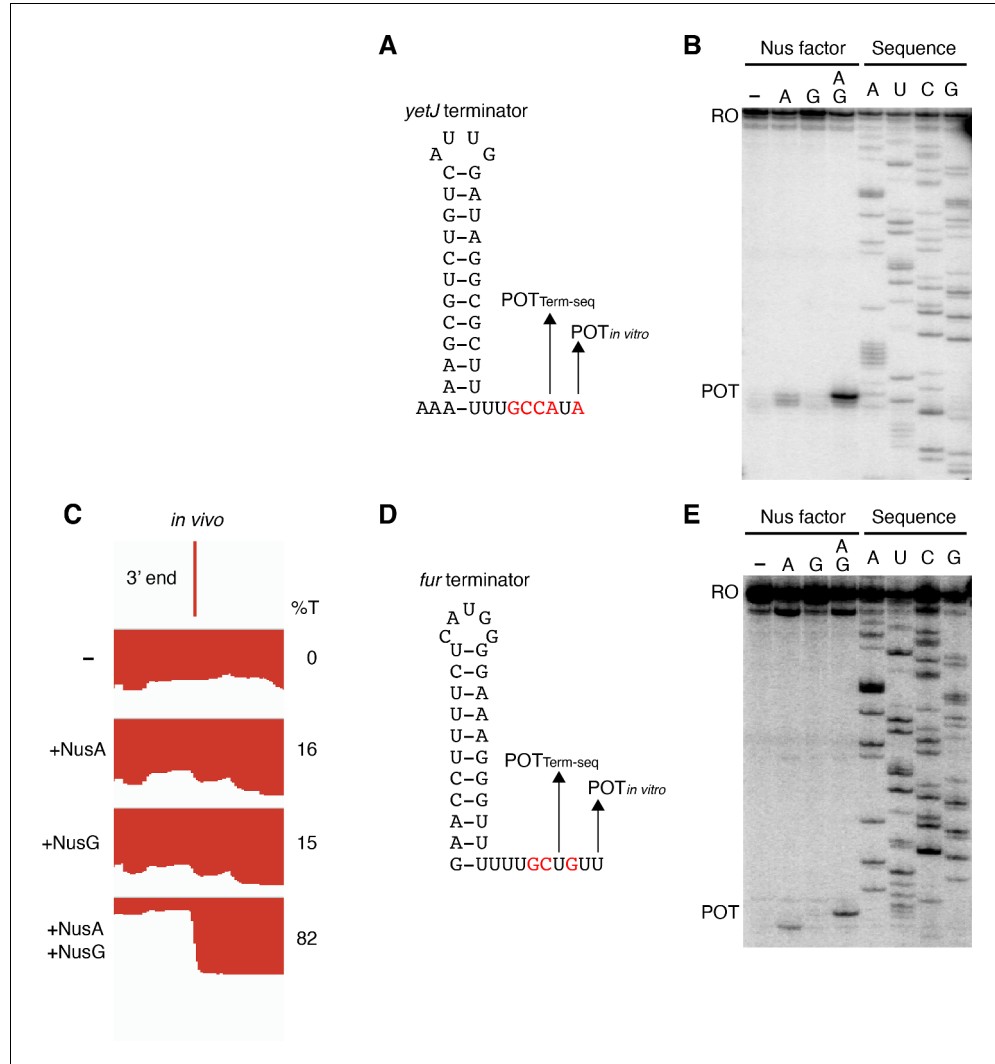

**Figure 4.** NusG stimulates terminators with particularly weak terminal base pairings. (**A**) *YetJ* terminator showing the point of termination identified in vivo by Term-seq (POT_Term-seq) and by in vitro transcription in the +A+G condition (POT_in vitro). Disruptions in the U-rich tract are shown in red. The upstream A tract is also shown. An IGV screenshot of this terminator is shown in *Figure 2A*. (**B**) Single-round in vitro termination assay with the *yetJ* terminator. Experiments were performed in the absence (–) or presence of NusA (**A**) and/or NusG (**G**) as indicated. Positions of terminated (POT) and run-off (RO) transcripts are marked. RNA sequencing lanes (A, U, C, G) are labeled. (**C**) IGV screenshot of a genomic window centered around the *fur* terminator. Top track is the 3' end identified by Term-seq. Bottom tracks are the RNA-seq coverage data for the *nusA*_dep Δ*nusG* (-), Δ*nusG* (+NusA), *nusA*_dep (+NusG), and WT (+NusA +NusG) strains. %T in each strain is shown on the right of each track. Transcription proceeds from right to left. (**D, E**) Identical to panels (**A, B**) except that it is the *fur* terminator. (**D**) Note that the terminal three base pairs contain the A tract and one G residue.

The online version of this article includes the following figure supplement(s) for figure 4:

**Figure supplement 1.** NusG stimulates termination at terminators containing A-U base pairs at the base of the hairpin.

**Figure supplement 2.** RNET-seq analysis of intrinsic termination.

**Figure supplement 3.** RNET-seq analysis of NusG-dependent pausing at intrinsic terminators.

**Figure supplement 4.** NusA is a more potent termination factor in vitro than NusG.

**Figure supplement 5.** Convergent transcription does not modify the impact of NusG in vitro.

**Figure supplement 6.** Comparison of Term-seq 3' ends, RNET-seq 3' ends, and in vitro 3' ends.

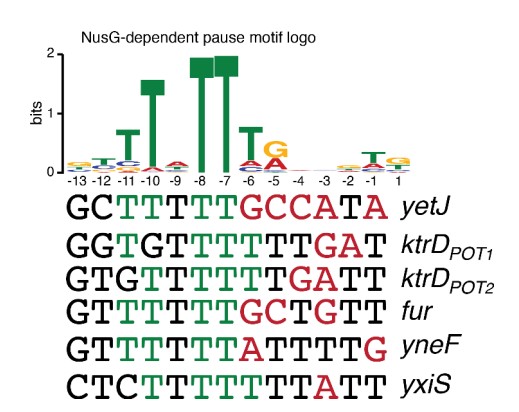

**Figure 5.** NusG-dependent pause motif. The NusG-dependent pause motif logo of the non-template DNA (ntDNA) strand is pictured on the top. ntDNA strand sequence upstream of the 3' end identified by in vitro transcription for NusG-dependent terminators identified in vivo. Green nucleotide (nt) are T residues that fit the NusG-dependent pause motif logo (TTNTTT). Red nt are non-U residues within the U-rich tract, which extends from positions −9 to −1.

been subjected to processing by 3'–5' exoribonucleases (*Yakhnin et al., 2020*; *Mondal et al., 2016*; *Dar and Sorek, 2018*).

Using our published RNET-seq data from WT *B. subtilis* (*Yakhnin et al., 2020*), we identified 485 peaks of nascent RNET-seq 3' ends located 0–4 nt downstream of the 3' end of an intrinsic terminator identified by Term-seq (*Supplementary file 4*). Notably, for 82% of these 3' ends, the RNET-seq 3' end was 1–4 nt downstream of the Term-seq 3' end (*Figure 4—figure supplement 2*). For each of these 485 intrinsic terminators, we revised the hairpin and U-rich tract compositions to reflect the position of the nascent 3' end identified by RNET-seq, instead of the released steady-state 3' end identified by Term-seq (*Supplementary file 4*).

Each NusG-dependent terminator tested in vitro contained several consecutive A-U/G-U base pairs at the bottom of the terminator hairpin stem. To establish whether this phenomenon occurred at a genome-wide level, we determined whether NusG-dependent terminators (Δ%T ≥ 25 in the Δ*nusG* strain) contained an enrichment of A-U/G-U base pairs at the bottom of the hairpin stems compared to terminators which were strong and independent of NusG (SI of NusG, %T ≥ 70 in the WT strain, 10 ≥ Δ%T ≥ −10 in the Δ*nusG* strain). Focusing on the revised terminators in *Supplementary file 4*, we tabulated the number of terminators that contained ≥1 consecutive terminal A-U/G-U base pairs at the bottom of the revised hairpin stem vs. those that had 0 terminal A-U/G-U base pairs. For these revised terminators, the SI of NusG terminators were directly compared to the NusG-dependent terminators via a Fisher's exact test, which revealed that NusG-dependent terminators were more likely to contain ≥1 consecutive terminal A-U/G-U base pairs than SI of NusG terminators (*Figure 4—figure supplement 2*). We also updated the set of all U-rich tracts that can be found in *Supplementary file 2* with the set of U-rich tracts found in *Supplementary file 4* to create a new total set of U-rich tracts encompassing all 1400 intrinsic terminators. Generating sequence logos for the SI of NusG and NusG-dependent terminators from this updated pool and comparing these logos using DiffLogo (*Nettling et al., 2015*) showed that NusG-dependent terminators exhibited a modest tendency to have distal U-rich tract interruptions compared to SI of NusG terminators (*Figure 4—figure supplement 2*).

While RNET-seq cannot by itself distinguish between pause sites that result in release vs. those that result in readthrough, by constraining the pause sites identified by RNET-seq to the positions of transcript release observed via Term-seq (*Supplementary file 4*), we were able to use RNET-seq to quantify pause strength at intrinsic terminators. The pause strength in the WT strain can be compared to the pause strength in the Δ*nusG* strain to determine the NusG dependency of a pause site (*Yakhnin et al., 2020*). Using this methodology, we determined the NusG dependency of pausing for each SI of NusG and NusG-dependent intrinsic terminator (*Figure 4—figure supplement 3*, *Supplementary file 4*). Interestingly, we found that strong NusG-dependent pausing could be found at both SI of NusG and NusG-dependent terminators. It should be noted that RNET-seq RNA isolation steps lead to a depletion of RNET-seq read coverage at intrinsic terminators. As such, we found that the intrinsic terminators identified by RNET-seq tended to have stronger RNA:DNA hybrids and stronger NusG-dependent pause signals compared to the total pool of intrinsic terminators, as these features stabilize RNAP at the POT. These findings complicated the direct comparison of RNET-seq data to Term-seq data.

## NusG and NusA cooperatively coordinate global gene expression

Terminator readthrough can impact gene expression by increasing transcription of downstream genes oriented in the same direction, destabilizing transcripts from downstream convergent transcription units due to formation of dsRNA and/or changing the expression of global regulators (*Mondal et al., 2016*). To examine the effect of NusG on gene expression, a differential expression analysis was conducted comparing expression data from each mutant strain to expression data from the WT strain (*Supplementary file 5*). Volcano plots were constructed based on the results of this analysis, and affected genes were determined using false discovery rate (FDR) cutoffs of 0.005 and fold change cutoffs of 4 (*Zhao et al., 2018*; *Dalman et al., 2012*). This approach revealed that NusG is involved in regulating gene expression on a global scale, with 106 transcripts increasing in expression and 37 transcripts decreasing in expression in the Δ*nusG* strain (*Figure 6A*). NusA had a larger effect on gene expression with 322 transcripts increasing in expression and 94 transcripts decreasing in expression in the *nusA*$_{dep}$ strain (*Figure 6B*). The cooperative relationship between those two proteins on intrinsic termination extended to gene expression, with 28% of all transcripts expressed in the WT strain being misregulated in the *nusA*$_{dep}$ Δ*nusG* strain (*Figure 6C and D*).

## NusG plays a critical role in regulating swimming motility

The transcripts per million (TPM) of all genes were calculated and the mutant TPM values were compared to their WT counterparts (*Supplementary file 6*). Through this analysis, it was found that loss of NusG resulted in a median threefold decrease in expression of genes within the motility regulon, while loss of both NusA and NusG resulted in a median eightfold decrease in expression of these genes (*Figure 7A*, *Supplementary file 7*). The 27 kb *fla/che* operon is composed of 32 genes involved in flagella biosynthesis and motility (*Cozy and Kearns, 2010*; *Márquez-Magaña and Chamberlin, 1994*; *Albertini et al., 1991*). Transcription of the *fla/che* operon is initiated by RNAP and the vegetative sigma factor σ$^A$, which in turn directs expression of the alternative sigma factor σ$^D$ encoded by the penultimate gene within the *fla/che* transcript, *sigD* (*Helmann et al., 1988*; *Márquez et al., 1990*; *Serizawa et al., 2004*; *Kearns and Losick, 2005*). To assess the expression trends of the *fla/che* operon, expression of the second gene of the cluster (*flgC*) and *sigD* was calculated, showing a 1.7-fold and threefold decrease in the expression of *flgC* and *sigD* in the Δ*nusG* strain, respectively, and a 2.1-fold and 10-fold decrease in the expression of *flgC* and *sigD* in the in the *nusA*$_{dep}$ Δ*nusG* strain, respectively (*Figure 7B and C*). As such, the ratio of *flgC* to *sigD* expression increased from 1.3-fold in the WT strain to 2.1-fold in the Δ*nusG* strain and 6.2-fold in the *nusA*$_{dep}$ Δ*nusG* strain (*Figure 7B and C*). Higher expression of the 5' portion of a transcript compared to the 3' portion of a transcript is sometimes caused by the activity of 3'–5' exoribonucleases such as

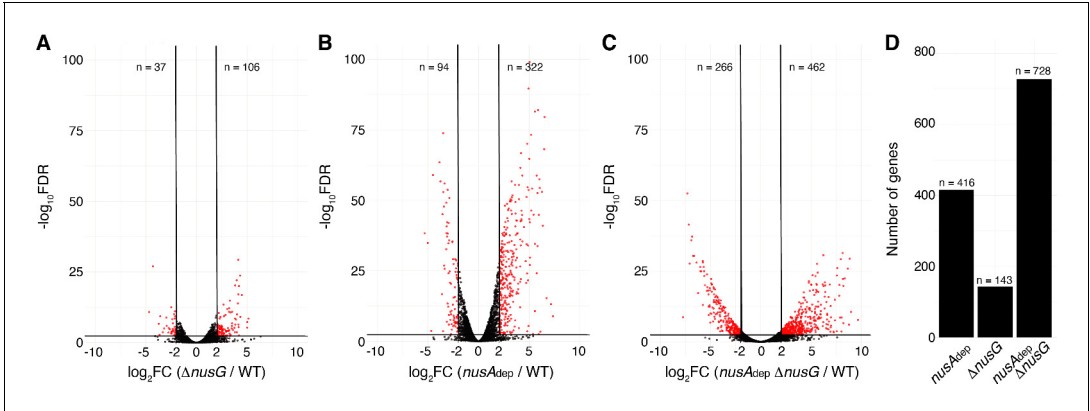

**Figure 6.** NusG coordinates global gene expression with NusA. (**A**) Volcano plot derived from differential expression analysis comparing steady-state gene expression levels in the wild-type (WT) and Δ*nusG* strains. Cutoffs are log$_2$ fold change (log$_2$FC) of 2 and a −log$_{10}$ of the false discovery rate (−log$_{10}$FDR) of 2.3. Number (n) of genes downregulated and upregulated are specified. (**B**) Identical to panel (**A**) except comparing the WT and *nusA*$_{dep}$ strains. (**C**) Identical to panel (**A**) except comparing the WT and *nusA*A$_{dep}$ Δ*nusG* strains. (**D**) Total number of differentially expressed genes in the *nusA*$_{dep}$, Δ*nusG*, and *nusA*$_{dep}$ Δ*nusG* strains; 2604 transcripts in which the transcripts per million (TPM) >10 in the WT strain were used in this analysis (*Supplementary file 6*).

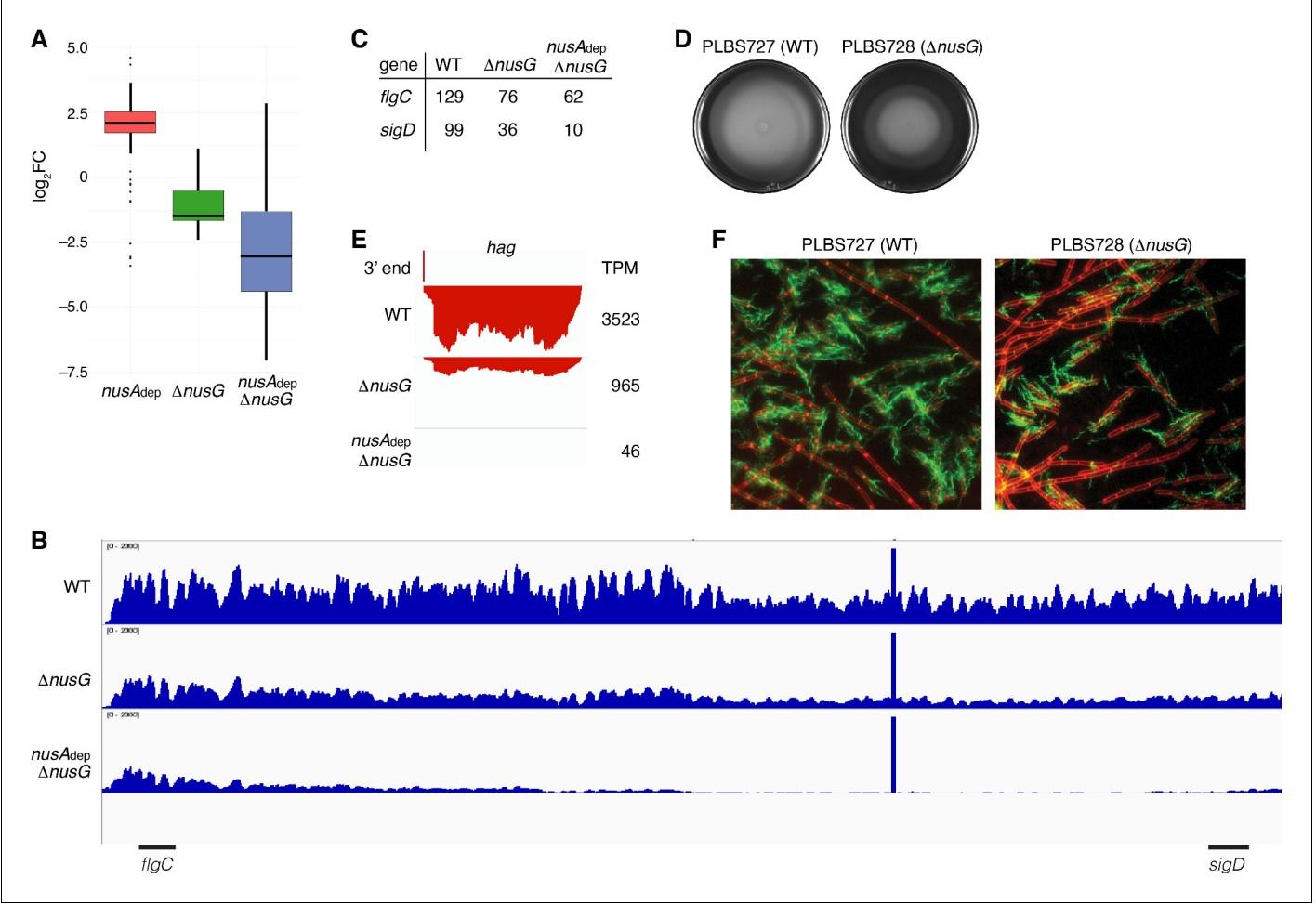

**Figure 7.** NusG is a motility factor in *Bacillus subtilis*. (**A**) Box plot showing the effect of NusA and/or NusG for all transcripts within the motility regulon; log$_2$ fold change (log$_2$FC) (mutant:WT [wild type]). (**B**) IGV screenshot of the *fla/che* operon. Each track is the RNA-seq coverage data for the WT, Δ*nusG*, and *nusA*$_{dep}$ Δ*nusG* strains. Locations of the *flgC* and *sigD* genes are specified below the screenshot. (**C**) Transcripts per million (TPM) values calculated for the *flgC* and *sigD* genes in the WT, Δ*nusG*, and *nusA*$_{dep}$ Δ*nusG* strains. (**D**) Swimming motility assay for PLBS727 (WT) and PLBS728 (Δ*nusG*). (**E**) IGV screenshot of the *hag* transcript. Top track is the 3' end identified by Term-seq. Bottom tracks are the RNA-seq coverage data for the WT, Δ*nusG*, and *nusA*$_{dep}$ Δ*nusG* strains. Transcription proceeds from right to left. TPM values were calculated for the *hag* transcript in each strain and are specified to the right of each track. (**F**) Fluorescence microscopy performed on PLBS727 (WT) and PLBS728 (Δ*nusG*) strains. Membrane is stained with FM4-64 (false colored red) and flagella are stained with Alexa Fluor 488 C$_5$ maleimide (false colored green).

The online version of this article includes the following figure supplement(s) for figure 7:

**Figure supplement 1.** NusG is a motility factor in *Bacillus subtilis*.

**Figure supplement 2.** NusA may serve as a transcription destabilization factor.

PNPase and RNase R (*Liu et al., 2014*; *Baumgardt et al., 2018*; *Bechhofer and Deutscher, 2019*). While there were moderate increases in the expression of the genes encoding for PNPase (*pnpA*) and RNase R (*rnr*) in the absence of NusG (*Figure 7—figure supplement 1*), the changes in expression of *pnpA* were not considered statistically significant during the differential expression analysis (*Supplementary file 5*), and the expression of *rnr* remained relatively low in all conditions (*Figure 7— figure supplement 1*). Thus, the observed increase in the ratio of *flgC* to *sigD* expression cannot be fully explained by an increase in known mediators of RNA decay, and is likely due to a defect in transcript completion, which has been posited to impact the expression of *sigD* compared to the 5' portion of the *fla/che* operon (*Cozy and Kearns, 2010*).

To explore how the changed expression of the *fla/che* transcript affected motility, a swimming motility assay was conducted on WT, *nusA*$_{dep}$, Δ*nusG*, and *nusA*$_{dep}$ Δ*nusG* strains (*Figure 7D*, *Figure 7—figure supplement 1*). To eliminate complications in the interpretation of the assay caused

by the differing growth rates of the NusA depleted strains, the assay was also conducted on strains PLBS727 (WT *nusG* and WT *nusA*) and PLBS728 (Δ*nusG* and WT *nusA*) (*Figure 7D*). Loss of NusG in our Δ*nusG* strain and in PLBS728 resulted in an impaired swimming motility phenotype. This impaired swimming motility phenotype may have been further exacerbated by depletion of NusA, although the reduced growth rate of the NusA depleted strains complicated interpretation (*Figure 7—figure supplement 1*).

The *hag* transcript, encoding the flagellin protein in *B. subtilis,* is one of the most abundant transcripts in the cell, and its expression is entirely dependent on σ$^D$-directed RNAP (*Mondal et al., 2016*; *LaVallie and Stahl, 1989*; *Mirel and Chamberlin, 1989*). Consistent with a cascading reduction in expression of the σ$^D$ regulon due to the reduction of *sigD* expression, expression of *hag* mRNA was reduced 4-fold in the Δ*nusG* strain and by 75-fold in the *nusA*$_{dep}$ Δ*nusG* strain (*Figure 7E*). To monitor how the reduction of *hag* expression impacts Hag production, we engineered a Hag variant with a surface exposed cysteine residue to be expressed ectopically from its native promoter. This Hag variant could then be fluorescently labeled via a cysteine-reactive fluorescent maleimide dye (*Blair et al., 2008*). Introducing this allele into the same six strains that were used for the swimming motility assay allowed us to determine whether the absence of NusA and/or NusG affected flagella synthesis levels (*Figure 7F*, *Figure 7—figure supplement 1*). Through this experiment, we observed both a decrease in the frequency of cells that produced flagella and a reduced number of flagella per cell. We conclude that the loss of NusG results in a motility defect that was correlated with a reduction in the expression of the *fla/che* operon, *hag*, and the majority of the σ$^D$ regulon.

## Discussion

### NusG is an intrinsic termination factor

In this work we conducted Term-seq in WT, *nusA*$_{dep}$, Δ*nusG*, and *nusA*$_{dep}$ Δ*nusG* strains of *B. subtilis.* By locating all intrinsic terminators in the WT condition, we were able to singly and combinatorially quantify the effect of NusA and NusG on intrinsic termination in vivo. We found that NusG stimulates termination at intrinsic terminators with suboptimal hairpins and strong NusG-dependent pause signals upstream of the 3′ end. We also found that NusG works in a cooperative fashion with NusA to regulate both intrinsic termination and global gene expression. This cooperativity during termination can be fully recapitulated by the NGN domain of NusG in vitro. While the AR2 domain of NusA and the NGN domain of NusG were found to physically interact in *E. coli* during elongation, this interaction is unlikely to be relevant to the regulation of elongation in *B. subtilis* due to the absence of the AR2 domain in *B. subtilis* NusA (*Strauß et al., 2016*). Thus, *B. subtilis* NusA and NusG likely work together without physical interaction to cooperatively stimulate termination in vivo. Only 12% of all intrinsic terminators identified in this study function independently of both NusA and NusG. Thus, our results establish that intrinsic (factor-independent) termination is primarily a factor-mediated process in *B. subtilis.* Thus, a new nomenclature system that divides intrinsic terminators based on their factor dependency profiles is warranted. We suggest NusA-dependent terminators, NusG-dependent terminators, NusA-NusG-dependent terminators, and factor-independent terminators.

In vitro experimentation was conducted on six terminators that were found to be stimulated by NusG in vivo, and it was generally found that NusA is the more potent termination factor in vitro (*Figure 2*, *Figure 4*, *Figure 4—figure supplement 1*, *Figure 4—figure supplement 4*). Interestingly, these templates included terminators that were found to be more reliant on NusG than NusA in vivo. This discrepancy could be explained by the inability of the in vitro assay to fully mimic in vivo conditions and/or the participation of additional protein factors in vivo. It was recently found that RNAP pausing at intrinsic terminators can lead to RNAP collision at convergent transcription units and that these collisions can result in transcript release within *E. coli* (*Ju et al., 2019*). Knowing that pausing is involved in NusG-dependent intrinsic termination, we tested the effect of convergent transcription in vitro on terminators that were both identified to be NusG-dependent and at which convergent transcription occurs in vivo. However, convergent transcription had no impact on the effect of NusG on termination in vitro (*Figure 4—figure supplement 5*).

Compared to the position identified by Term-seq in vivo, we found that NusG stimulated termination at a position further downstream in three templates tested in vitro (*Figure 2C*, *Figure 4B and*

*E*). This finding can be explained by the fact that NusG shifts RNAP to the post-translocation register in both *E. coli* and *B. subtilis* (*Sevostyanova and Artsimovitch, 2010*; *Yakhnin et al., 2020*), and may thus function as a processivity factor in *B. subtilis* until encountering the consensus pause motif at these terminators. NusA has been reported to stimulate termination at a more upstream position (*Yakhnin and Babitzke, 2002*). This phenomenon was only observed in vitro in reactions where NusG was absent, implying that the shifting of RNAP to the post-translocation register is a central feature of elongation complexes containing NusG. Moreover, NusA and NusG together stimulated termination in vitro at a position 1–3 nt further downstream than the 3' end identified by Term-seq in vivo on all templates tested (*Figure 3B*, *Figure 4B and E*, *Figure 4—figure supplement 1*). Our RNET-seq analysis also showed that the 3' ends of nascent transcripts are consistently downstream of the steady-state 3' ends of released transcripts, although the RNET-seq 3' ends did not always correspond precisely to the in vitro POT(s) (*Supplementary file 4*, *Figure 4—figure supplement 6*). The discrepancy between the in vivo released POT(s) and in vivo nascent POT(s) can likely be attributed to exoribonuclease trimming of the terminated transcripts in vivo, which has shown to occur via YhaM-mediated degradation in the related firmicute *Streptococcus pyogenes* (*Lécrivain et al., 2018*).

## NusG-dependent pausing at terminators with terminal A-U/G-U base pairing is a critical component of NusG-dependent termination

Our in vitro results establish that NusG-dependent pausing is a critical component of NusG-dependent termination. We found that NusG is able to elicit several consecutive pauses at the *ktrD* terminator, only some of which result in termination (*Figure 3A*). A feature that distinguishes the *ktrD* NusG-dependent terminator from the *ktrD* hairpin-stabilized NusG-dependent pause is whether the hairpin can extend to within 7–9 nt of the RNA 3' end (*Figure 3B*). The NusG-dependent pause likely extends the time frame for terminator hairpin completion before readthrough occurs. This pause is sequence-specific and occurs when RNAP incorporates U residues into the nascent transcript at positions −12, −11, −8, −7, and −6 relative to the 3' end at position −1. These U residues correspond to the TTNTTT consensus pause motif in the ntDNA strand of the paused transcription bubble that was identified previously (*Figure 5*; *Yakhnin et al., 2016*; *Yakhnin et al., 2020*). Note that the U residues at positions −12 and −11 would always be present in the base of the terminator hairpin.

Terminator hairpins with weak terminal base pairs are highly atypical due to the various structural elements of RNAP that stabilize the elongation complex, including the interactions between the Sw3 pocket and the lid of RNAP with the −10 and −9 positions of the RNA transcript, respectively (*Ray-Soni et al., 2016*). Displacement of these interactions requires major energetic expenditures, which may explain why certain bacterial species evolved a strong G-C preference at the base of the terminator hairpin to drive completion of its formation (*Ray-Soni et al., 2016*; *Peters et al., 2011*). Weak RNA hairpins with several consecutive terminal A-U/G-U base pair and distal U-rich tract interruptions were a feature of all NusG-dependent terminators mapped in vitro, and our analysis of RNET-seq data showed that these characteristics are enriched across NusG-dependent terminators genome-wide (*Figure 3*, *Figure 4*, *Figure 4—figure supplement 1*, *Figure 4—figure supplement 2*, *Figure 5*).

## Model of NusA-dependent and NusG-dependent intrinsic termination

The precise mechanism of transcript release at Nus-dependent terminators is not currently understood. One potential mechanism is that NusA assists with the formation of suboptimal terminator hairpins, while NusG-dependent pausing provides additional time for hairpin completion, which drives transcript release in all cases where the U-rich tract is at or above an optimality threshold. Additionally, the ability of NusG to shift RNAP into the post-translocation register may assist in the hyper-translocation and deactivation of RNAP at intrinsic terminators with weak terminal base pairs and distal U-tract interruptions, while NusA assists in the formation of suboptimal hairpins and/or pausing at suboptimal U-rich tracts. The observation that NusA stimulates termination at a more upstream position provides evidence for a model in which terminators that depend on the formation of terminal A-U/G-U base pairs require NusG, while terminators that have G-C or C-G terminal base pairs may not be able to induce a sufficiently stable NusG-dependent pause, and therefore will depend solely on NusA (*Figure 8*). Moreover, our data suggests that the main feature that

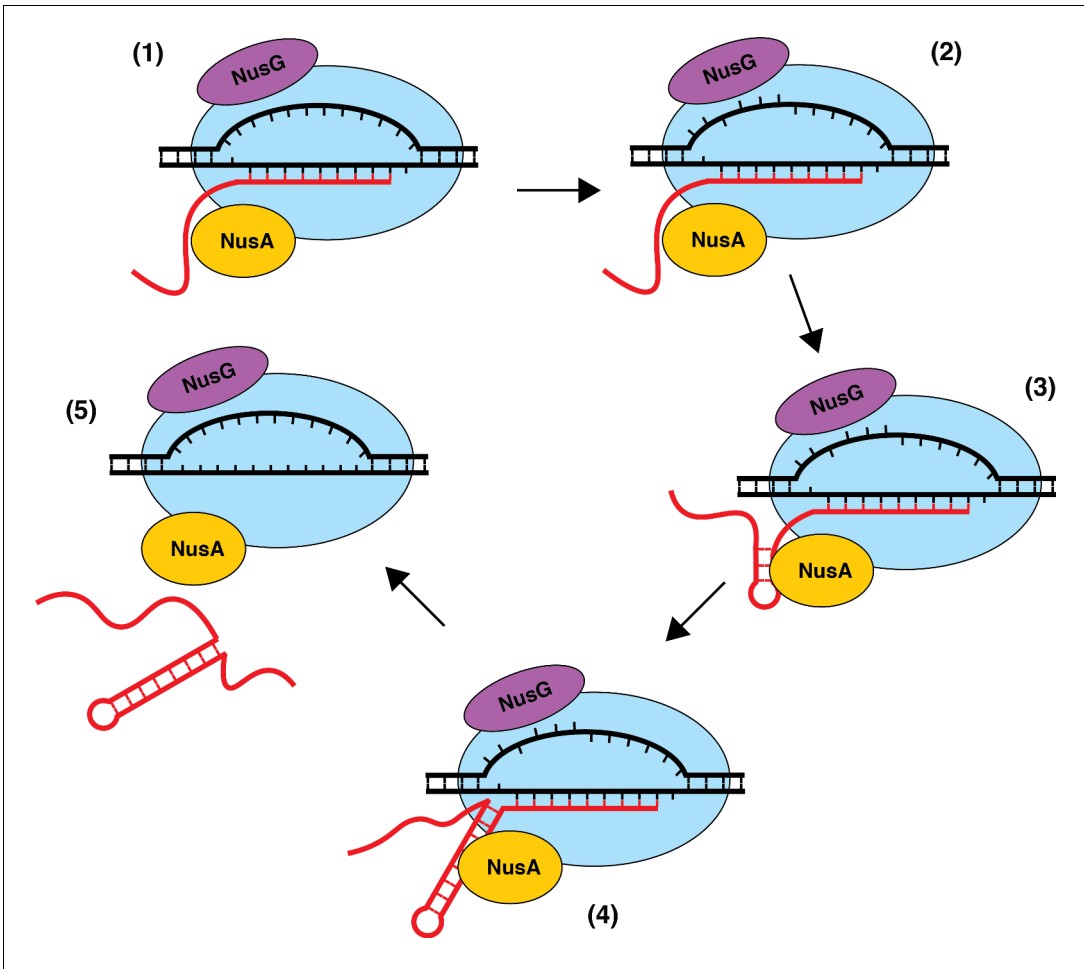

**Figure 8.** Model of NusA-dependent and NusG-dependent intrinsic termination. (1) During transcription elongation, NusA binds to the β flap domain of RNA polymerase (RNAP) near the RNA exit channel, whereas NusG binds to the β' clamp helices in close proximity to the non-template DNA (ntDNA) strand within the transcription bubble. NusA stimulates pausing during transcription of a suboptimal U-rich tract containing distal U tract interruptions. (2) NusG shifts RNAP to the post-translocated state and increases the pause half-life by making sequence-specific contacts with the TTNTTT motif within the ntDNA strand of the paused transcription bubble. The T residues in this motif are shown as being flipped out toward the NGN domain of bound NusG, but evidence for base flipping has not been obtained. (3) NusA assists with hairpin formation within the RNA exit channel. (4) NusG-dependent pausing provides time for weak A-U and/or G-U base pairs to form at the base of the terminator hairpin such that the hairpin to 3' end distance is reduced to 7–9 nt. (5) The combination of the weak RNA-DNA hybrid, the close proximity of the hairpin to the RNA 3' end, and the long-lived pause contributes to transcript release.

differentiates terminators that require NusA and NusG (Req A and G) from terminators that require either NusA or NusG (Req A or G) is hairpin strength, with Req A and G terminators having significantly weaker hairpins than Req A or G terminators (*Figure 1C*, *Supplementary file 3*). This model suggests that the most suboptimal terminators (Req A and G) require both the NusG-dependent pause and the hairpin stimulation activity of NusA, while the comparatively more optimal (Req A or G) terminators can function efficiently with only one of these activities.

## NusG is required for normal cellular motility

Loss of NusG results in an impaired motility phenotype that can be traced back to a changed pattern in expression of the *fla/che* operon. A decrease in the expression of the 3' portion of the *fla/che* operon compared to the 5' portion was previously reported in *B. subtilis* (*Cozy and Kearns, 2010*). The possibility that NusG functions as a processivity factor in *B. subtilis* may explain this defect in

transcript completion. Moreover, the fact that NusA depletion alone resulted in an increase in expression across the entire *fla/che* operon is suggestive that NusA may serve as an anti-processivity factor (*Figure 7—figure supplement 2*). The increase in expression of the motility regulon in the *nusA*$_{dep}$ strain can likely be explained by an increase in the expression of *sigD* (*Figure 7A*, *Figure 7—figure supplement 2*; *Estacio et al., 1998*; *West et al., 2000*). These results hint that modulating the activity of NusG and/or NusA governs the frequency of completion of the *fla/che* operon transcript and serves as a method for *B. subtilis* to regulate the switch between motile and sessile states.

## NusG-dependent termination might be a conserved mechanism

*B. subtilis* NusG contacts the ntDNA strand to elicit a pause via an NT dipeptide located within the NGN domain (*Yakhnin et al., 2016*). A phylogenetic analysis focusing on bacterial species that contain a *B. subtilis*-like NusG homolog with an NT or HT dipeptide shows that the ability of NusG to contact the ntDNA strand may be present in a large number of Gram-positive and Gram-negative phyla (*Figure 9*; *Yakhnin et al., 2020*). Moreover, mycobacterial NusG was found to contain an NT dipeptide at this position and thus the discovery that mycobacterial NusG stimulates intrinsic termination at suboptimal terminators in vitro (*Czyz et al., 2014*) suggests that the NusG-dependent termination mechanism may be conserved as well.

# Materials and methods

**Key resources table**

| Reagent type (species) or resource | Designation | Source or reference | Identifiers | Additional information |
|---|---|---|---|---|
| Strain, strain background (*Bacillus subtilis*) | MH5636 | *Qi and Hulett, 1998* | N/A | See *Supplementary file 8* for derivatives |
| Strain, strain background (*Bacillus subtilis*) | PLBS338 | *Yakhnin et al., 2004* | N/A | See *Supplementary file 8* for derivatives |
| Strain, strain background (*Bacillus subtilis*) | 3610 | *LaVallie and Stahl, 1989* | N/A | See *Supplementary file 8* for derivatives |
| Antibody | Rabbit polyclonal anti-NusA | Peter Lewis | N/A | WB (1:5000) |
| Antibody | Rabbit polyclonal anti-SigA | Masaya Fujita | N/A | WB (1:5000) |
| Antibody | Goat polyclonal peroxidase labeled anti-rabbit | GenScript | Cat# A00098 | WB (1:1000) |
| Recombinant DNA reagent | pTZ19R | Thermo Fisher | Cat# SD0141 | See *Supplementary file 9* for derivatives |
| Recombinant DNA reagent | pNC018 | David Rudner | N/A | See *Supplementary file 9* for derivatives |
| Recombinant DNA reagent | pNE4 | *Blair et al., 2008* | N/A | See *Supplementary file 9* for derivatives |
| Sequence-based reagent | Primers | This study | N/A | See *Supplementary file 10* |
| Software, algorithm | Python | N/A | v.3.83 | https://www.python.org/ |
| Software, algorithm | R | N/A | v.4.03 | https://www.r-project.org/ |
| Software, algorithm | ImageQuant | N/A | v.5.2 | N/A |

*Continued on next page*

*Continued*

| Reagent type (species) or resource | Designation | Source or reference | Identifiers | Additional information |
|---|---|---|---|---|
| Software, algorithm | TransTermHP | *Kingsford et al., 2007* | v.2.09.1 | http://transterm.ccb.jhu.edu/ |
| Software, algorithm | DiffLogo | *Nettling et al., 2015* | v.3.12 | https://bioconductor.org/packages/release/bioc/html/DiffLogo.html |
| Software, algorithm | DESeq2 | *Love et al., 2014* | v.1.26.0 | https://bioconductor.org/packages/release/bioc/html/DESeq2.html |
| Software, algorithm | Kallisto | *Bray et al., 2016* | v.0.46.2 | https://pachterlab.github.io/kallisto/ |
| Software, algorithm | Trimmomatic | *Bolger et al., 2014* | v.0.38 | http://www.usadellab.org/cms/?page=trimmomatic |
| Software, algorithm | Cutadapt | *Marcel, 2011* | v.1.16 | https://cutadapt.readthedocs.io/en/stable/ |
| Software, algorithm | Bedtools | *Quinlan and Hall, 2010* | v.2.26.0 | https://bedtools.readthedocs.io/en/latest/ |
| Software, algorithm | Samtools | *Li et al., 2009* | v.0.1.19–44428 cd | http://www.htslib.org/ |
| Software, algorithm | bwa-mem | *Li, 2013* | v.0.7.12-r1034 | http://bio-bwa.sourceforge.net/bwa.shtml |
| Software, algorithm | BLASTp | *Altschul et al., 1990* | v.2.11 webtool | https://blast.ncbi.nlm.nih.gov/Blast.cgi |
| Software, algorithm | iTOL | *Letunic and Bork, 2016* | v.5 webtool | https://itol.embl.de/ |
| Software, algorithm | NCBI taxonomic tree webtool | *Sayers et al., 2009* | webtool | https://www.ncbi.nlm.nih.gov/guide/howto/gen-com-tree/ |
| Software, algorithm | MEME | *Bailey and Elkan, 1994* | v.4.12.0 | https://meme-suite.org/meme/tools/meme |
| Software, algorithm | BaseSpace | Illumina | N/A | https://basespace.illumina.com/dashboard |
| Software, algorithm | BioPython | *Cock et al., 2009* | v.1.77 | https://biopython.org/ |
| Software, algorithm | numpy | *Harris et al., 2020* | v.1.19.0 | https://numpy.org/ |
| Software, algorithm | scipy | *Virtanen et al., 2020* | v.1.5.1 | https://www.scipy.org/ |
| Software, algorithm | RNAStructure | *Reuter and Mathews, 2010* | v.6.0.1 | https://rna.urmc.rochester.edu/RNAstructureWeb/Servers/Predict1/Predict1.html |
| Software, algorithm | ggplot2 | *Wickham, 2016* | v.3.2.1 | https://ggplot2.tidyverse.org/ |
| Software, algorithm | IGV | *Robinson et al., 2011* | v.2.4.14 | http://software.broadinstitute.org/software/igv/ |
| Software, algorithm | Term-seq peak calling pipeline | 'Term-seq' Github repository | N/A | https://github.com/zfmandell/Term-seq |

*Continued on next page*

*Continued*

| Reagent type (species) or resource | Designation | Source or reference | Identifiers | Additional information |
|---|---|---|---|---|
| Other | CIP | NEB | (Discontinued) | Term-seq |
| Other | Ribo-zero | Illumina | (Discontinued) | Term-seq |
| Other | T$_4$ RNA Ligase 1 | NEB | Cat# M0204S | Term-seq |
| Other | α-32$^P$ UTP | PerkinElmer | Cat# BLU007H25OUC | Urea-PAGE |
| Other | Agarose | Dot Scientific | Cat # DSA20090-50 | Agarose gels |
| Other | Polyacrylamide | Fisher Sci | Cat# HBGR337500 | Urea-PAGE |
| Other | FM-64 | Invitrogen | Cat# T13320 | Fluorescence Microscopy |
| Other | Maleimide dye | Invitrogen | Cat# A10254 | Fluorescence Microscopy |
| Other | RNeasy columns | Qiagen | Cat# 74106 | Term-seq |
| Other | Lysozyme | Sigma-Aldrich | Cat# L6876 | Term-seq |
| Other | RNA sequencing reagents | Illumina | Cat# 20020594 | Term-seq |
| Other | Urea | EMD | Cat# 666122–2.5 kg | Urea-PAGE |
| Other | PVDF membrane | Thermo | Cat# 88585 | Western blot |
| Other | ECL substrate | Thermo Fisher | Cat# 32209 | Western blot |
| Chemical compound, drug | IPTG | Dot Scientific | Cat# DSI56000-25 | Cell Culture – NusA production |
| Chemical compound, drug | Chloramphenicol | Sigma-Aldrich | Cat# C0378-25 g | Cell Culture |
| Peptide, recombinant protein | NusA | This study | N/A | 1 μM |
| Peptide, recombinant protein | NusG | This study | N/A | 1 μM |
| Peptide, recombinant protein | NGN only NusG | This study | N/A | 1 μM |
| Peptide, recombinant protein | Mutant NGN only NusG | This study | N/A | 1 μM |
| Peptide, recombinant protein | RNAP core | This study | N/A | 0.19 μM |
| Peptide, recombinant protein | SigA | This study | N/A | 0.38 μM |

## *B. subtilis* strains

All strains used in this study are listed in *Supplementary file 8*. Strains PLBS727 and PLBS728 are WT and Δ*nusG* strains, respectively. PLBS730 is a NusA depletion strain. This strain contains an IPTG-inducible *nusA* allele and *E. coli lacI* integrated into the chromosomal *amyE* gene (*Yakhnin et al., 2020*). PLBS731 is identical to PLBS730 except that it also contains Δ*nusG* (*Yakhnin et al., 2020*). NusA production was maintained in these two strains by culturing cells in the presence of 0.2 mM IPTG. PLBS730 and PLBS731 grown + 0.2 mM IPTG were considered to be WT and Δ*nusG*, respectively. These strains grown in the absence of IPTG were considered as *nusA*$_{dep}$ and *nusA*$_{dep}$ Δ*nusG*, respectively. The *lacA::P$_{hag}$-hag$^{T209C}$ tet* construct was generated by digesting P$_{hag}$-hag$^{T209C}$ fragment from plasmid pNE4 using BamHI and SphI (*Konkol et al., 2013*). The digested fragment was ligated into the BamHI and SphI sites of pNC018 (*lacA::tet*) to generate plasmid pKB141 (*Konkol et al., 2013*). Plasmid pKB141 was introduced into strain DS2569 by natural

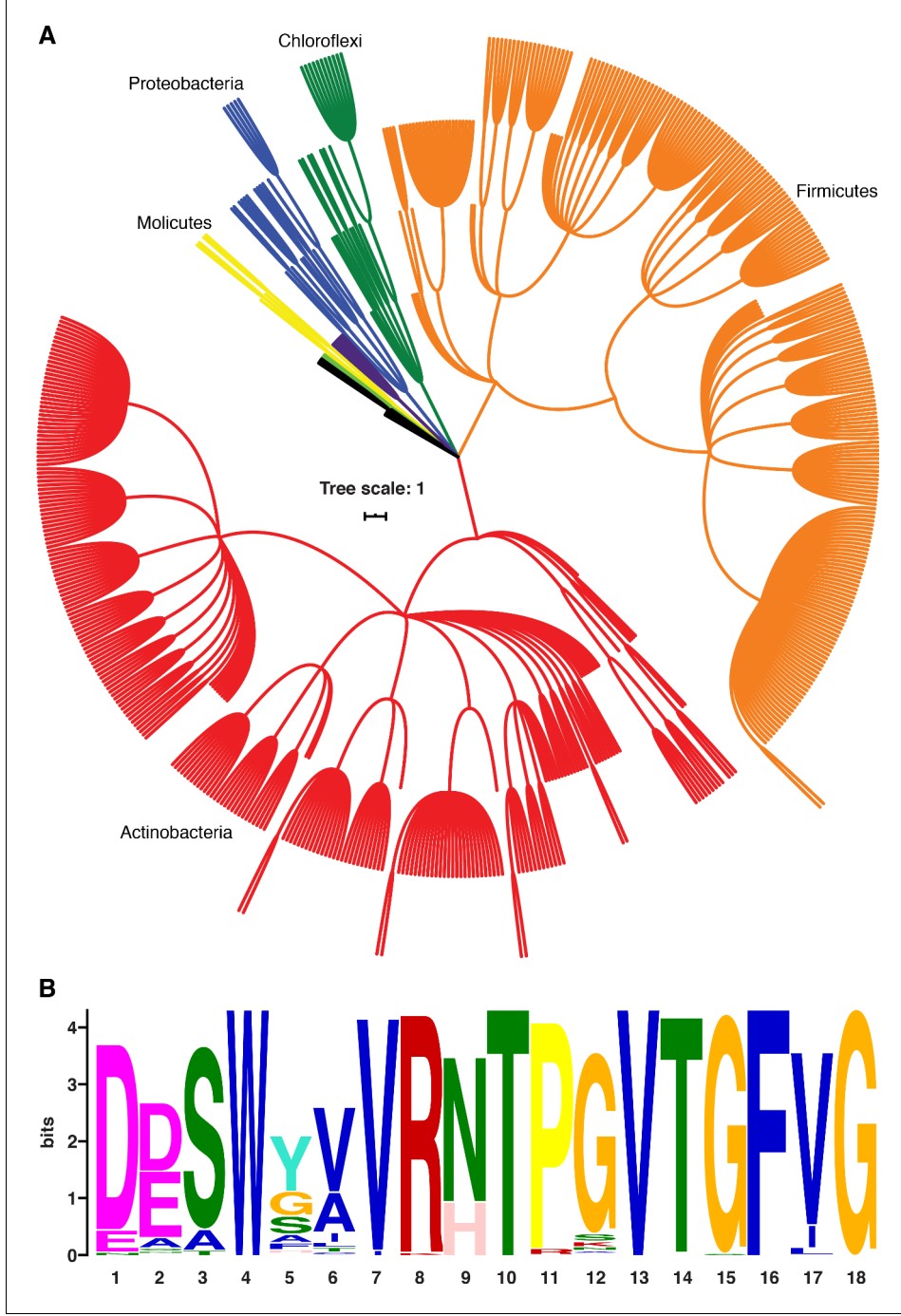

**Figure 9.** NusG homologs exhibit the capacity to contact the ntDNA strand across the bacterial domain. (**A**) Phylogenetic tree constructed from the 16S rRNA sequences of all bacteria found to encode a NusG homolog that contains the *Bacillus subtilis*-like dipeptide residues (NT and HT). Each different phylum of bacteria that is represented with three or more species is highlighted a different color. Actinobacteria branches are red, Firmicute branches are orange, Chloroflexi branches are dark green, Proteobacteria branches are blue, Synergistaceae branches are purple, Mollicute branches are yellow, and Bacteroidete branches are light green. Phyla with fewer than three representative species are in black. (**B**) Sequence logo constructed from the portion of NusG that interacts with the ntDNA strand in *B. subtilis* for all NusG homologs present in (**A**). Critical dipeptide NT/HT is located at positions 9 and 10.

competence to generate strain DS6331 and further introduced into appropriate strain backgrounds by SPP1-mediated transduction (*Konkol et al., 2013*). All bacterial strains and plasmids are available from the corresponding author.

### *B. subtilis* growth and library generation

Each strain was streaked onto LB plates containing 20 µg/mL chloramphenicol and 0.2 mM IPTG. Single colonies of each strain were grown at 37°C as overnight standing cultures in 5 mL of LB media supplemented with 0.4 mM IPTG and 20 µg/mL chloramphenicol. The next day, 1 mL of cells were collected and washed twice with LB media. For strains to be depleted of NusA, a 30-fold dilution was made into 25 mL of LB supplemented with 20 µg/mL chloramphenicol (no IPTG). For strains in which NusA expression was maintained, a 75-fold dilution was made into 25 mL of LB media supplemented with 20 µg/mL chloramphenicol and 0.2 mM IPTG. All cultures were grown shaking at 37°C and both total RNA and total protein were extracted during mid-exponential phase. Barcoded Illumina libraries were generated from oligo-ligated transcripts as described previously (*Mondal et al., 2016*). Total RNA was CIP-treated and rRNA was depleted from each sample using ribo-zero rRNA depletion kits (Illumina). The remaining transcripts were then ligated to a unique 2′,3′-dideoxy RNA oligonucleotide (IDT) that was phosphorylated on the 5′ end. TruSeq standard mRNA libraries were generated from these samples. Equal amounts of each library were pooled and 150 nt single-read sequencing was performed with an Illumina NextSeq 500 in High Output mode.

### Western blot

NusA depletion was confirmed via Western blot for all replicates (*Figure 1—figure supplement 1*). Protein samples (3 µg) were fractionated in a 10% SDS gel and transferred to a 0.2 µm PVDF membrane. Purified His-tagged NusA, $\sigma^A$, and cell lysates were probed with rabbit anti-NusA or anti-$\sigma^A$ antibodies (1:5000 dilution), and developed using enhanced chemiluminescence following incubation with HRP-conjugated goat anti-rabbit antibody (GenScript). Two images were taken of each membrane, one after probing for NusA and a second after probing for $\sigma^A$.

### Data processing, analysis, and identification of 3′ ends

Illumina sequencing generated 141,842,192 reads across eight samples (WT, $nusA_{dep}$, $\Delta nusG$, and $nusA_{dep}$ $\Delta nusG$). All reads were processed to comprehensively yield all 3′ ends as described previously with modifications (*Mondal et al., 2016*). After demultiplexing, Illumina adapters were trimmed with Trimmomatic, resulting in a traditional RNA-seq dataset (*Bolger et al., 2014*). Cutadapt was then used to extract all reads that were found to contain the unique RNA oligonucleotide used during library generation, resulting in a Term-seq dataset (*Marcel, 2011*), which was mapped to the *B. subtilis* 168 genome (NC_000964.3) via bwa-mem in single-end mode (*Li, 2013*). Bam files for each pair of replicates were merged, the contents of each resulting bam file was split by strand using samtools, and coverage files were generated for each strand-specific bam file using bedtools (*Li et al., 2009*; *Quinlan and Hall, 2010*). A series of custom python scripts were used to comprehensively identify all 3′ ends by calculating the coverage variation ($C_V$) at each nt of all strand-specific coverage files, and identifying the local maxima across these $C_V$ landscapes as described previously (*Mondal et al., 2016*). The $C_V$ magnitude at a 3′ end is tightly correlated with 3′ abundance, which is a function of transcript abundance, 3′ end stability, and termination efficiency (%T) in cases where the 3′ end is a result of termination (*Mondal et al., 2016*). To limit 3′ ends that could be attributed to noise, a $C_V$ threshold was set at $\geq 10$ for this study. All strand-specific 3′ ends were then merged by biological condition, thereby generating the final 3′ end bedgraph files that contained the genomic location, the $C_V$, and the strand information of each identified 3′ end (*Supplementary file 1*). RNA-seq coverage files were generated by aligning the merged RNA-seq datasets to the *B. subtilis* genome using bwa-mem in single-end mode, splitting the resultant bam files by strand using samtools, using bedtools to calculate the per-nucleotide RNA-seq coverage, which were then merged by biological condition (*Li, 2013*; *Li et al., 2009*; *Quinlan and Hall, 2010*).

For construction of the phylogenetic tree, 10,000 NusG homologs were identified by querying the NusG recognition region (DDSWXXVR<u>XX</u>PXVXGFXG) using BLASTp, where X indicates any amino acid and the underlined region is the dipeptide by which *B. subtilis* NusG uses to contact the ntDNA strand (*Yakhnin et al., 2016*; *Altschul et al., 1990*). From these 10,000 homologs, 776

representative bacterial genera were identified, 617 of which were found to contain a *B. subtilis*-like dipeptide within the underlined region (NT or HT), and these species were chosen to construct a 16S rRNA-based phylogeny using the NCBI common taxonomy tree webtool (*Sayers et al., 2009*). Tree annotation and display were created with the interactive tree of life web platform (iTOL) (*Letunic and Bork, 2016*). All sequence logos used in this study were generated using the MEME suite and compared using DiffLogo (*Bailey and Elkan, 1994*; *Nettling et al., 2015*).

In silico intrinsic terminator prediction was conducted for *B. subtilis* via TransTermHP (*Kingsford et al., 2007*). To ascertain the magnitude of overlap between *B. subtilis* terminators identified in this study, our previous study (*Mondal et al., 2016*), and by TransTermHP (*Kingsford et al., 2007*), we compared the strand identity and 3′ end location of all identified terminators in each population. We considered a terminator to be matched between two Term-seq populations in cases where the strand identity was identical and the 3′ end position of the terminator was within a 4 nt window in both datasets, and matched between a Term-seq population and a TransTermHP population when the strand identity was identical and the 3′ end position of the terminator was within a 15 nt window in both datasets.

## Differential expression analysis and replicate reproducibility

All RNA-seq reads post-trimming were pseudo-mapped using Kallisto in SE mode using the `-rf-stranded` option to a transcriptome built with Illumina-generated RNA-seq data collected from strain PLBS338 (*Bray et al., 2016*; *Ritchey et al., 2020*). This method determined both TPM and raw count values for each annotated transcript for both merged and non-merged FASTQ files (*Supplementary file 6*). TPM values for all coding sequence-containing transcripts were compared for each pair of replicates via both a scatter plot and a Spearman's correlation analysis (*Figure 1—figure supplement 5*). All replicates were found to be highly reproducible with a mean Spearman's r-value of 0.964. Genes that were differentially expressed between the WT and each mutant strain were identified by analyzing the raw count data from each strain via DESeq2 (*Supplementary file 5*; *Figures 6A, B and C*; *Love et al., 2014*). A variance stabilizing transformation was applied to the transcriptome-wide raw count data for each strain, and this matrix was projected onto 2D space via a principal component analysis (PCA). This analysis revealed that transcriptome-wide expression data collected from each sample clustered neatly by strain (*Figure 1—figure supplement 5*). To ensure that the Δ%T values provided in this work were not due to noise, total %T values from each replicate were compared using pairwise Spearman's correlation analyses. These pairwise r-values were then organized into a correlational matrix plot (*Figure 1—figure supplement 5*).

## Terminator screening and characterization

A 3′ end can be the result of intrinsic termination, Rho-dependent termination, or RNA decay (*Roberts, 2019*). An intrinsic terminator contains a GC-rich RNA hairpin and a U-rich tract immediately downstream of the hairpin (*Roberts, 2019*). Some intrinsic terminators also contain an A-rich tract upstream of the hairpin (*Roberts, 2019*). As such, the 50 nt upstream of each 3′ end was iteratively sent through an in silico RNA secondary structure prediction algorithm (RNAStructure) (*Mathews et al., 2004*). In cases where a hairpin was identified, the presence of a U tract leading to the 3′ end was verified by visual inspection. At least 2 U residues were considered to be a viable U-rich tract, as long as the Us were appropriately positioned and consecutive as seen at the *liaH* terminator, which had the U-rich tract UUCCGCACG (*Supplementary file 2*). This was the only intrinsic terminator with a U-rich tract with only 2 Us. Formation of the final two base pairs of a terminator hairpin requires the greatest energetic expenditure and are the rate limiting steps of terminator hairpin completion (*Ray-Soni et al., 2016*). As such, bacterial systems have evolved a heavy GC preference at these positions (*Ray-Soni et al., 2016*; *Peters et al., 2011*). Once the hairpin has completed, termination occurs 7–9 nt downstream from the terminal hairpin nt (*Mondal et al., 2016*; *Ray-Soni et al., 2016*). These details were factored into intrinsic terminator prediction by assuming that the U-rich tract started with the first U residue for each identified terminator.

Termination efficiency (%T) of a particular intrinsic terminator can be calculated by comparing the median RNA-seq coverage value of the 10 nt upstream (U) to the median RNA-seq coverage value of the 10 nt downstream (D) of the identified 3′ end using the following equation: %T = [(U−D)/U]* (100) as described previously (*Mondal et al., 2016*). Short window sizes of 10 nt were chosen to limit

potential complications arising from transcription initiation downstream of the POT. A 3' end containing the intrinsic terminator modules was included in this study only in cases where %T $\geq$ 5 in the WT strain. For each intrinsic terminator 3' end identified in our WT strain, where x is the genomic coordinate of the intrinsic terminator 3' end, we searched for a corresponding 3' end identified in our previous study within a [x−3, x+3] window (*Mondal et al., 2016*). In cases where a corresponding 3' end was identified, the %T provided in this study was calculated from the position in which the 3' end was identified previously (*Mondal et al., 2016*). We chose this approach to maintain the genomic coordinates of the 3' ends from the prior study. We found that for the majority of matched 3' ends, the distance between the previously identified and currently identified 3' ends was 0–1 nt. For all newly identified 3' ends, the %T was calculated based on the position of the newly identified 3' end.

To determine the effect of an elongation factor, or combination of elongation factors, on the %T of a particular intrinsic terminator, one can calculate the change in termination efficiency (Δ%T) using the following equation: $\Delta\%T = \%T_{WT} - \%T_{mutant}$. This approach was systematically applied to all intrinsic terminators to determine the general effect of an elongation factor, or combination of elongation factors, on intrinsic termination (*Supplementary file 2*). Determination of all terminator hairpin stem lengths and loop lengths was derived from the result of sending the predicted terminator hairpin stem through the in silico RNA secondary structure prediction algorithm (RNAStructure) (*Mathews et al., 2004*).

## RNET-seq data analysis

Total lists of RNET-seq 3' ends were obtained as outlined previously (*Yakhnin et al., 2020*). The RNET-seq 3' ends that could be attributed to intrinsic termination were obtained as follows. For each Term-seq intrinsic terminator 3' end, where x is the genomic coordinate of the 3' end, the total number of RNET-seq 3' ends at each nucleotide within the window [x,x+4] (proximal to the intrinsic terminator 3' end) and within the window [x−150,x+4] (upstream of the intrinsic terminator 3' end) was tabulated. A 3' end was considered to be potentially due to intrinsic termination in cases where both an RNET-seq 3' end was identified within the proximal window, and the number of 3' ends identified in the upstream window passed a coverage threshold in which the 75th percentile of 3' end abundances across this window was >0. Much like for Term-seq intrinsic terminator identification, the RNET-seq 3' end assigned as the intrinsic terminator was the most abundant 3' end found within the proximal window (*Supplementary file 4*).

Normalized RNET-seq 3' end abundance were calculated as the 3' abundance at the identified intrinsic terminator, divided by the 75th percentile of 3' end abundance across the upstream window. An intrinsic terminator was only included in the analysis if the normalized 3' end abundance was greater than the 25th percentile of all normalized RNET-seq intrinsic terminator abundances. The NusG dependency of a 3' end was calculated as the $\log_2$ transformed ratio of the WT RNET-seq normalized 3' end abundance, divided by the Δ*nusG* RNET-seq normalized 3' end abundance. To ensure that intrinsic terminators with no corresponding 3' ends in the Δ*nusG* strain were included in the analysis, the normalized abundance value for these terminator 3' ends in the Δ*nusG* strain was set at 0.01.

## DNA templates and plasmids

All pAY196 derivatives were generated using a strategy akin to site-directed mutagenesis PCR (*Hemsley et al., 1989*). The entirety of pAY196 was amplified using Vent polymerase (New England Biolabs) using outward-directed primer pairs with constant sequences complementary to the plasmid backbone and flanking regions containing the biological sequence of interest as described previously (*Mondal et al., 2016*). To prevent the formation of primer dimers or internal hairpins caused by terminator hairpins, the biological sequence of one primer contained an A tract when present and the 5' portion of the predicted hairpin ending at the 3' most nt of the loop. The biological sequence of the other primer contained the 3' portion of the predicted hairpin and 19 nt downstream of the predicted hairpin. All plasmids and primers used in this study are listed in *Supplementary file 9* and *Supplementary file 10*, respectively.

## In vitro transcription

Analysis of RNAP pausing and termination was performed as described previously with modifications (*Mondal et al., 2017*). DNA templates were PCR-amplified from plasmids containing either WT or mutant terminator sequence, both of which included the predicted terminator hairpin, 19 nt downstream of the predicted hairpin, and the A tract when present, fused to the *B. subtilis* $P_{trp}$ promoter and *trp* leader-derived C-less cassette (pAY196 and derivatives) using PSL (modifies the $P_{trp}$ promoter to a consensus promoter with an extended −10 element) and lacZ primers (*Figure 2—figure supplement 1*). Halted elongation complexes containing a 27 nt transcript were formed for 5 min at 37°C by combining equal volumes of 2× template (50–200 nM) with 2× halted elongation complex master mix containing 80 µM ATP and GTP, 2 µM UTP, 100 µg/mL bovine serum albumin, 150 µg/mL (0.38 µM) *B. subtilis* RNAP holoenzyme, 0.76 µM SigA, 2 µCi of [α-$^{32}$P]UTP and 2× transcription buffer (1× = 40 mM Tris-HCl, pH 8.0, 5 mM $MgCl_2$, 5% trehalose, 0.1 mM EDTA, and 4 mM dithiothreitol). RNAP and SigA were added from a 20x stock solution containing 1.5 mg/mL RNAP and 0.35 mg/mL SigA in enzyme dilution buffer (1× = 20 mM Tris-HCl, pH 8.0, 40 mM KCl, 1 mM dithiothreitol, and 50% glycerol). A 4× solution containing either 0 µM NusA and 0 µM NusG, 4 µM NusA and 0 µM NusG, 0 µM NusA and 4 µM NusG, or 4 µM NusA and 4 µM NusG in 1× transcription buffer was added, and the resulting solution was incubated for 5 min at 23°C. For termination assays, a 4× extension master mix containing 80 µM KCl, 600 µM of each NTP, 400 µM rifampicin, in 1× transcription buffer was added, and the reaction was allowed to proceed for 30 min at 23°C before the addition of an equal volume of 2× stop/gel loading solution (40 mM Tris-base, 20 mM $Na_2$EDTA, 0.2% sodium dodecyl sulfate, 0.05% bromophenol blue, and 0.05% xylene cyanol in formamide). For pausing assays, the same extension master mix was added, and the reaction was incubated at 23°C, with aliquots removed and stop/gel loading solution added at the specified time points. A 30 min time point was included for all pausing assays that mirrored the experimental conditions of the termination assay. RNA bands were separated on standard 5% sequencing polyacrylamide gels. All RNA sequencing reactions were conducted like other termination reactions, albeit with the addition of one of four 3' dNTPs at a 1:1 molar ratio with the corresponding NTP within the extension master mix. Termination efficiencies and pausing half-lives were quantified as described previously (*Yakhnin and Babitzke, 2002*). Each in vitro experiment was conducted a minimum of two times, with representative gels shown. Values for pause half-lives are averages ± standard deviation.

## Convergent transcription in vitro

gBlock gene fragments (IDT) containing biological sequence (100 nt upstream of, 50 nt downstream of the predicted *serA* and *fisB* terminator hairpins) flanked by two identical 27 nt C-less cassettes, inward-facing consensus promoters with extended −10 elements ($P_{forward}$ and $P_{reverse}$), and EcoRI and HindII restriction digestion sites (NEB) were cloned into pTZ19R (Thermo Fisher). DNA templates containing both $P_{forward}$ and $P_{reverse}$ were PCR-amplified from the appropriate pTZ19R derivative using a primer pair that was specific for pTZ19R (M13_2.0 and M13_reverse_2.0, IDT, derivatives of M13 and M13 reverse universal sequencing primers). Conversely, DNA templates containing just $P_{forward}$ were PCR-amplified from the appropriate pTZ19R derivative using a reverse primer specific for pTZ19R (M13_2.0) and a forward primer specific for the biological sequence (serA_uni/fisB_uni) (*Figure 2—figure supplement 1*). In vitro transcription termination reactions were conducted identically using both of the templates detailed in *Figure 2—figure supplement 1*.

## Motility assay

Swimming motility assays were conducted as performed previously with modifications (*Mukherjee et al., 2016*). Strains were grown to mid-exponential phase in the presence of 0.2 mM IPTG and concentrated to 10 $OD_{600}$ in phosphate-buffered saline (PBS) pH 7.4 (137 mM NaCl, 2.7 mM KCl, 10 mM $Na_2HPO_4$, and 2 mM $KH_2PO_4$). LB plates containing 0.3% Bacto agar ± 0.2 mM IPTG were dried for 10 min in a laminar flow hood, centrally inoculated with 10 µL of the cell suspension, dried for another 10 min, and incubated for ~13 hr at 37°C in a humid chamber. Plates were visualized with a BioRad Geldoc system and digitally captured using BioRad Quantity One software.

## Microscopy

Fluorescence microscopy was conducted with a Nikon 80i microscope with a phase contract objective Nikon Plan Apo 100× and an Excite 120 metal halide lamp as described previously with modifications (*Mukherjee et al., 2016*). FM4-64 (Molecular Probes) was visualized with a C-FL HYQ Texas Red Filter Cube (excitation filter 532–587 nm, barrier filter >590 nm). Alexa 488 $C_5$ maleimide (Molecular Probes) fluorescent signals were visualized using a C-FL HYQ FITC Filter Cube (FITC, excitation filter 460–500 nm, barrier filter 515–550 nm).

To visualize flagella, cells were grown at 37°C in LB broth + 0.2 mM IPTG to mid-exponential phase. One mL of broth culture was harvested and resuspended in 50 µL of PBS containing 5 µg/mL Alexa Fluor 488 $C_5$ maleimide (Molecular Probes) and incubated for 3 min at 23°C as described previously (*Blair et al., 2008*). Cells were then washed with 1 mL of PBS. Membranes were stained by resuspension of 30 µL of PBS containing 5 µg/mL FM4-64 and incubated for 5 min at 23°C, then washed with 1 mL of PBS. Cell pellets were resuspended with 30–50 µL of PBS, and then 4 µL of suspension were placed on a microscope slide and immobilized with a poly-L-lysine treated coverslip. For cells to be depleted of NusA (PLBS730 and PLBS731), cells were initially grown in LB supplemented with 0.2 mM IPTG to mid-exponential phase. One mL of culture was harvested, washed 2× with fresh LB, back diluted in fresh LB, and grown for four generations at 37°C. One mL of culture was harvested and staining of the flagella and membrane was conducted via the above protocol.

## Code availability

All custom scripts used for 3' end mapping are available at https://github.com/zfmandell/Term-seq (*Mandell, 2020*; copy archived at swh:1:rev:48c039c50c1932aed66d8a423293bae6be66488c) and all other scripts are available upon request.

## Acknowledgements

Illumina sequencing was performed at the Penn State Genomics Core Facility. This work was supported by National Institutes of Health Grant GM098399 to Paul Babitzke, National Institutes of Health Grant GM131783 to Daniel B Kearns, and the Intramural Research Program of the National Institutes of Health/National Cancer Institute to Mikhail Kashlev.

## Additional information

### Funding

| Funder | Grant reference number | Author |
| --- | --- | --- |
| National Institutes of Health | GM098399 | Paul Babitzke |
| National Institutes of Health | GM131783 | Daniel B Kearns |
| National Institutes of Health | intramural | Mikhail Kashlev |

The funders had no role in study design, data collection and interpretation, or the decision to submit the work for publication.

### Author contributions

Zachary F Mandell, Data curation, Software, Formal analysis, Investigation, Methodology, Writing - original draft, Writing - review and editing; Reid T Oshiro, Alexander V Yakhnin, Investigation, Writing - review and editing; Rishi Vishwakarma, Investigation; Mikhail Kashlev, Daniel B Kearns, Supervision, Funding acquisition, Writing - review and editing; Paul Babitzke, Conceptualization, Formal analysis, Supervision, Funding acquisition, Writing - original draft, Project administration, Writing - review and editing

### Author ORCIDs

Alexander V Yakhnin http://orcid.org/0000-0002-7313-7054
Paul Babitzke https://orcid.org/0000-0003-2481-1062

Decision letter and Author response
Decision letter https://doi.org/10.7554/eLife.61880.sa1
Author response https://doi.org/10.7554/eLife.61880.sa2

# Additional files

## Supplementary files

• Supplementary file 1. 3' ends identified in each strain. Column A (3' end), genomic coordinate of 3' end identified by Term-seq. Column B (strand), strand information of identified 3' end. Column C (Cv), the coverage variation (Cv) calculated at the genomic coordinate of the identified 3' end. Column D (3' end and upstream 50 nucleotide [nt]), 3' end position and 50 nt upstream of 3' end (51 nt total) of the ntDNA strand. Column E ($\Delta$G-51nt (kcal/mol)), the free energy of RNA folding obtained from the 50 nt upstream of the 3' end and the 3' end. Column F (%T), the termination efficiency calculated at the identified 3' end. Column G (relative position), the transcriptomic context of the 3' end.

• Supplementary file 2. All intrinsic terminators identified by Term-seq in the wild-type (WT) strain. Column A (point of termination [POT]), genomic coordinate of the identified intrinsic terminator. Column B (strand), strand information of the identified intrinsic terminator. Column C (relative position), distance of identified intrinsic terminator from nearest upstream coding sequence and identity of coding sequence. Column D (%T [WT]), termination efficiency of identified intrinsic terminator in the WT strain. Column E (%T [mutant]), termination efficiency of identified intrinsic terminator in the mutant strain. Column F ($\Delta$%T), change in termination efficiency due to loss of factor(s). Column G (upstream sequence), nucleotide composition of the 9 nt upstream of the predicted terminator hairpin. Column H (predicted hairpin sequence), nucleotide composition of the predicted terminator hairpin. Column I (downstream sequence), nucleotide composition of the 9 nt downstream of the predicted terminator hairpin. Column J ($\Delta$G-hairpin [kcal/mol]), the free energy of RNA folding the predicted terminator hairpin.

• Supplementary file 3. Pairwise p-values.

• Supplementary file 4. All intrinsic terminators identified by RNET-seq in the wild-type (WT) strain. Column A (Term-seq point of termination [POT]), genomic coordinate of the intrinsic terminator identified by Term-seq. Column B (strand), strand information of the identified intrinsic terminator. Column C (dist), the distance in nucleotide (nt) of the RNET-seq 3' end from the Term-seq 3' end. Column D (relative position), distance of the intrinsic terminator identified by RNET-seq from nearest upstream coding sequence and identity of coding sequence. Column E (revised upstream sequence), nucleotide composition of the 9 nt upstream of the predicted terminator hairpin revised using the RNET-seq 3' end. Column F (revised predicted hairpin sequence), nucleotide composition of the predicted terminator hairpin revised using the RNET-seq 3' end. Column G (revised downstream sequence), nucleotide composition of the 9 nt downstream of the predicted terminator hairpin revised using the RNET-seq 3' end. Column H (classification), whether the terminators were classified as SI of NusG or NusG-dependent. Column I (log$_2$ fold change [log$_2$FC]), the log$_2$ transformed ratio of the pause strength in the WT strain compared to the $\Delta$nusG strain.

• Supplementary file 5. Results of differential expression analysis. Column A (transcript), transcript, naming system = ILL (transcriptome was built using Illumina sequencing data), official gene symbol, Locus, location of coding sequence within transcript. Column B (baseMean), normalized counts for transcript in column A, averaged across all samples. Column C (log$_2$ fold change), the log$_2$ transformation of the fold change in expression of the transcript in column A. Column D (lfcSE), standard error of the log$_2$(fold change) calculated in column C. Column E (stat), Wald statistic for log$_2$(fold change) calculated in column C. Column F (p-value), Wald p-value for log$_2$(fold change) calculated in column C. Column G (p$_{adj}$), Benjamni-Hochberg p-adjusted value for log$_2$(fold change) calculated in column C.

• Supplementary file 6. Transcriptome-wide expression data collected for all strains as reported in transcripts per million (TPM) and raw counts. Column A (transcript), transcript, naming system = ILL (transcriptome was built using Illumina sequencing data), official gene symbol, Locus, location of

coding sequence within transcript. Column B (WT Rep. 1), data collected from wild-type (WT) replicate 1. Column C (WT Rep. 2), data collected from WT replicate 2. Column D (WT Merged), data collected from merged WT replicates. Column E ($nusA_{dep}$ Rep. 1), data collected from $nusA_{dep}$ replicate 1. Column F ($nusA_{dep}$ Rep. 2), data collected from $nusA_{dep}$ replicate 2. Column G ($nusA_{dep}$ Merged), data collected from merged $nusA_{dep}$ replicates. Column H (Δ$nusG$ Rep. 1), data collected from Δ$nusG$ replicate 1. Column I (Δ$nusG$ Rep. 2), data collected from Δ$nusG$ replicate 2. Column J (Δ$nusG$ Merged), data collected from merged Δ$nusG$ replicates. Column K ($nusA_{dep}$ Δ$nusG$ Rep. 1), data collected from $nusA_{dep}$ Δ$nusG$ replicate 1. Column L ($nusA_{dep}$ Δ$nusG$ Rep. 2), data collected from $nusA_{dep}$ Δ$nusG$ replicate 2. Column M ($nusA_{dep}$ Δ$nusG$ Merged), data collected from merged $nusA_{dep}$ Δ$nusG$ replicates.

• Supplementary file 7. Expression data for the motility regulon. Column A (transcript), transcript, naming system = ILL (transcriptome was built using Illumina sequencing data), official gene symbol, Locus, location of coding sequence within transcript. Column B (wild-type), normalized TPM values collected from merged wild-type replicates for the transcript specified in column A. Column C ($nusA_{dep}$), normalized transcripts per million (TPM) values collected from merged $nusA_{dep}$ replicates for the transcript specified in column A. Column D (Δ$nusG$), normalized TPM values collected from merged Δ$nusG$ replicates for the transcript specified in column A. Column E ($nusA_{dep}$ Δ$nusG$), normalized TPM values collected from merged $nusA_{dep}$ Δ$nusG$ replicates for the transcript specified in column A.

• Supplementary file 8. *Bacillus subtilis* strains used in this study.
• Supplementary file 9. Plasmids used in this study.
• Supplementary file 10. Primers used in this study.
• Transparent reporting form

## Data availability

RNA-seq data were deposited in GEO under accession number GSE154522. All other data generated or analysed during this study are included in the manuscript and supporting files.

The following dataset was generated:

| Author(s) | Year | Dataset title | Dataset URL | Database and Identifier |
|---|---|---|---|---|
| Mandell ZF, Oshiro RT, Yakhnin AV, Vishwakarma R, Kashlev M, Kearns DB, Babitzke P | 2021 | NusG is an intrinsic transcription termination factor that stimulates motility and coordinates gene expression with NusA | https://www.ncbi.nlm.nih.gov/geo/query/acc.cgi?acc=GSE154522 | NCBI Gene Expression Omnibus, GSE154522 |

The following previously published dataset was used:

| Author(s) | Year | Dataset title | Dataset URL | Database and Identifier |
|---|---|---|---|---|
| Yakhnin AV, FitzGerald PC, McIntosh C, Yakhnin H, Kireeva M, Turek-Herman J, Mandell ZF, Kashlev M, Babitzke P | 2020 | NusG Controls Transcription Pausing and RNA Polymerase Translocation Throughout the Bacillus subtilis Genome | https://www.ncbi.nlm.nih.gov/sra/PRJNA603835 | NCBI Sequence Read Archive, PRJNA603835 |

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
