## [Decision Letter]

**Acceptance summary:**

NusG is a transcription elongation factor conserved across all domains of life. This paper shows that NusG in the bacterium *Bacillus subtilis* promotes transcription termination of hundreds of RNAs, the first demonstration of NusG participating in this process. This work demonstrates that transcription termination mechanisms vary between bacterial species, and identifies another role for a ubiquitous transcription factor.

**Decision letter after peer review:**

Thank you for submitting your article "NusG is an intrinsic transcription termination factor that stimulates motility and coordinates gene expression with NusA" for consideration by *eLife*. Your article has been reviewed by 3 peer reviewers, including Joseph T Wade as the Reviewing Editor and Reviewer #1, and the evaluation has been overseen by Gisela Storz as the Senior Editor.

The reviewers have discussed the reviews with one another, and the Reviewing Editor has drafted this decision to help you prepare a revised submission.

Summary:

The reviewers and editors were enthusiastic about the major conclusion of the study: that NusG-dependent pausing is an important factor that promotes Rho-independent transcription termination in *Bacillus subtilis*. Nonetheless, we felt that this conclusion can be strengthened with additional analysis and experiments that hopefully are not terribly burdensome. We believe that these additions would bring the paper to the level required for publication in *eLife*. The essential revisions are detailed below, followed by the individual reviews.

Essential revisions:

1. While we appreciated the careful follow-up work, we felt that the major conclusion could be strengthened by a more in-depth analysis of the genome-wide data, assuming those data support the role of NusG-dependent pausing in termination. Reviewers 1 and 2 give specific suggestions in their reviews. Some of these relate to the way comparisons are made between datasets, and others address specific scientific questions. Of particular relevance are analyses that test whether NusG stimulates termination at sites with (i) weak terminal base-pairs, and (ii) gaps in the U-tract. Any other analyses of the genome-wide data that support the importance of NusG-simulated pausing in termination would be valuable to include. For example, is there any evidence that NusG-dependent pause sites identified by NET-seq are associated with sites of termination?

2. Termination sites in vitro are consistently downstream of those observed in vivo. While it is reasonable to hypothesize that this difference is due to trimming by exonucleases, there is no experimental evidence presented to support this. To test the hypothesis, we suggest mapping termination sites by 3' RACE in RNase mutant strains for one or two of the terminators characterized in the paper. B subtilis 3' exonucleases are defined, and mutant strains have been described (e.g., Oussenko et al., 2005 J Bacteriol 187:2758; Liu et al., 2014 Mol Microbiol 94:41).

3. Add a figure showing the model described in the discussion (lines 332-84) for the proposed roles of NusG and NusA in intrinsic termination.

4. Broaden the discussion of how the study relates to prior work on intrinsic termination, as detailed by Reviewer 2.

Reviewer #1:

This manuscript focuses on the role of NusA and NusG in intrinsic transcription termination in *Bacillus subtilis*. The authors argue that NusA and NusG are required for efficient termination at many sites, with NusG promoting termination by stimulating pausing. Stimulation of pausing by NusG appears to facilitate termination at sequences that would not ordinarily be expected to function as terminators. Lastly, the authors show that loss of NusA and/or NusG leads to large-scale changes in gene expression, including changes in expression of flagellar genes that lead to the loss of motility in a nusG deletion.

Finding a widespread role for NusG in intrinsic termination is interesting, and the authors make a reasonably strong case that this effect is due to NusG-dependent pausing. Nonetheless, several of the arguments made would be strengthened by further analysis of the genome-wide datasets. The effects of losing NusA/G on gene expression are somewhat less interesting; these changes are not surprising for such important regulators. Moreover, the physiological significance of these changes is unclear. Do NusG levels fluctuate sufficiently for there to be changes in gene expression? I think it is unlikely that NusG levels would ever drop sufficiently to impact motility.

There are a lot of comparisons of distributions in the manuscript. For the most part, the differences are clear, but a statistical analysis is warranted in the less obvious cases, e.g. Supplementary Figure 3.

Figure 4 shows a few examples of NusG stimulating termination at sites with weak terminal base-pairs. This is an interesting result, but with only a few examples it is difficult to judge how widespread this is. The authors should analyze their genome-wide datasets to determine if this is a common feature of NusG-dependent terminators.

Line 225. The authors argue that NusG stimulates termination at sites with gaps in the U-tract. This is an intriguing hypothesis, but should be tested using the genome-wide data.

Reviewer #2:

Mandell et al. report results of a genome-scale analysis of transcriptional termination in wild-type *Bacillus subtilis* and derivative strains in which NusA, NusG, or both factors are depleted or deleted. The work is of particular significance because effects of NusG on stimulating termination, while known, have not been characterized mechanistically. The authors report that NusG promotes termination at intrinsic terminators with suboptimal features, and often does so additively with effects of NusA. They propose that NusG enhances intrinsic termination via stimulation of a sequence-specific pause, although only a correlation rather than a causal relationship for this effect on pausing and effects on termination is established. Finally, the authors link the loss of NusG to motility defects. Overall, this comprehensive analysis of terminators in Bacillus is highly informative, and the analysis of the NusA/NusG-dependence of these terminators provides important insights into transcription regulatory mechanisms. Nonetheless, the manuscript could be improved by addressing the following main points.

1. The authors demonstrate a correlational but not causal relationship between NusG stimulation of pausing and its effect to enhance termination. They show that the pause exists without termination, but do not directly link pausing to termination. Have the authors identified mutations of NusG that still interact with RNAP but are defective for pause stimulation? Was a mutant NusG defective for the NT-strand interaction tested for the ability to promote termination? Unless more supportive data can be provided, the claim of causality should be tempered and the need for testing causality should be described.

2. Section starting at line 162 and Figure 3. Related to point 1, the authors perform a detailed analysis of the ktrD terminator and focus on possible effects of NusG on pausing downstream from the site of termination mapped by term-seq (A+8 numbered from the last G-C base pair in the terminator hairpin). The authors conclude that NusG exerts its effect on intrinsic termination through its role in pausing (line 177), but do not highlight in the Results the fact that they do not observe pausing or termination at A+8 in vitro. If the effects of NusG on pausing all occur after the point of termination in vivo, then how could pauses at those locations possibly affect termination in vivo? The authors should address this question explicitly in the Results, since it seems to contradict their conclusion. The same point appears to apply to the yetJ and fur terminators (Figure 4). In the Discussion the authors suggest the discrepancy could be related to nuclease trimming in vivo of RNAs terminated at the sites seen in vitro. Readers will already be confused by the discrepancy in the Results, so it would better to bring up this idea, which is reasonable, in the Results. Have the authors considered testing it, however? Can nuclease mutants be used to see if pausing and termination really correlate in vivo and in vitro? Here again, unless a more definitive test is performed, the conclusions should highlight where uncertainty remains and be framed around the need for additional experiments.

3. The comparisons of different classes of terminators (stimulated by NusA, NusG, both, or neither; Figures 1, S3, S8) are reported using box plots or sequence logos without any accompanying statistical analysis for the significance of differences or similarities of the characteristics catalogued. Additionally, the box plots are ineffective in illustrating differences in distributions relative to the more information-rich violin plots or ridge plots that could be generated for the same datasets (e.g., using the R function geom_violin from the R package ggplot2 and the R function geom_density_ridges from the R package ggridges). Box plots work best when data are unimodal and roughly symmetric, which these data are not.

Additionally, when making statements about comparisons of characteristics between large datasets, it is accepted practice to include an estimate of the statistical significance of any differences described (i.e., a p value, although more sophisticated estimates of statistical significance are also possible). The authors should consider alternative presentations distributions (e.g., of %T) and should include estimates of statistical significance for any comparison from which they draw conclusions. For example, the change in %T from WT could easily be tested for statistical significance using a Wilcoxon signed-rank test (a non-parametric paired difference test that avoids distributional assumptions). The distribution of predicted terminator hairpin stem lengths could be evaluated using a permutation test of shuffled classes.

4. The manuscript could be improved by placing the results in the broader context of what is already known and reported in the literature about intrinsic termination, NusA, and NusG. Currently, the authors' description of prior knowledge is more solipsistic than comprehensive. This narrow perspective limits in the impact of the work and at some points is misleading about what is already known or in debate. Examples of omissions are listed here. Correcting them comprehensively would greatly improve the manuscript.

(a) The authors do not cite a major prior study of genome-scale intrinsic termination in *B. subtilis* (Lalanne et al., 2018 Cell 173:749). They should compare their results to those of Lalanne et al. at multiple places in the manuscript. In validating their term-seq data (lines 109-115), the authors should include a comparison of their findings to those of Lalanne et al. In the analysis of misregulation by NusA/NusG depletion/deletion, it seems obvious to ask whether NusA, NusG, or both contribute significantly to the 167 intergenic, 'tuned' intrinsic terminators reported by Lalanne et al. to singly or in combination set the expression of 276 genes in *B. subtilis*. To what extent do contributions of NusA and NusG to termination at these 167 sites explain the misregulation of gene expression reported in the current manuscript?

(b) (line 96) As best I can tell, term-seq was developed independently by the Babitzke group and by the Sorek group in Israel (Dar et al., 2016 Science 352:aad9822). However, Dar et al. are not cited when describing the method. It would be appropriate to cite both the authors' own work and Dar et al. at this point in the manuscript.

(c) At multiple places in the manuscript (abstract, line72, line 314), the authors suggest that current or general dogma is that intrinsic terminators are not affected by transcription factors (or perhaps they mean not significantly affected by transcription factors). These statements employ a common trope in the scientific literature to frame and oversimplify prior knowledge so as to increase the perceived novelty of findings, and often constitute a strawman in scientific rhetoric. In this case, stating that intrinsic terminators not being significantly affected by transcription factors is a general dogma ignores long-standing knowledge that NusA stimulates intrinsic termination. Indeed NusA was discovered as an intrinsic termination factor and has long been described as such. See, for example, Greenblatt et al., 1980 PNAS 77:1991 "L factor that is required for β-galactosidase synthesis is the nusA gene product involved in transcription termination"; Schmidt and Chamberlin, 1987 JMB 195:809 "nusA protein of *Escherichia coli* is an efficient transcription termination factor for certain terminator sites"; Kingston and Chamberlin, 1981 Cell 27:523, which states in the abstract "termination at [rrnB] tL is dependent on the nusA protein"; Farnham et al., 1982 Cell 29:945 "Effects of NusA protein on transcription termination in the tryptophan operon of *Escherichia coli*" or, more recently, Gusarov and Nudler, 2001 Cell 107:437 "Control of intrinsic transcription termination by N and NusA: the basic mechanisms." Thus, there is no dogma that intrinsic terminators are not affected by transcription factors. There may be confusion about the definition of intrinsic termination, which is by no means exclusive to this manuscript. Intrinsic terminators are terminators that do not require transcription factors to exhibit at least some level of termination; they are not defined as terminators that are not affected by transcription factors. It is accurate to say the characteristics of intrinsic terminators that are affected by NusA (or in this case NusA and NusG) are not well characterized. The authors' results make a major contribution to such characterization, but they should be presented in that context and rather than the tired trope of overthrowing a dogma that doesn't really exist.

(d) At multiple places in the manuscript (line 15, line 72, line 302), the authors suggest that NusG was not previously known to affect transcriptional termination. This statement also is misleading. Mycobacterial NusG is known to stimulate intrinsic termination by Mycobacterial RNA polymerase and to stimulate intrinsic termination additively with NusA at some terminators (Czyz et al., 2016 mBio 5: e00931). The authors should include this precedent when introducing NusG effects on termination.

(e) (line 334) The authors note that "NusG shifts RNAP to the post-translocation register". This was previously reported (Sevostyanova and Artsimovitch, 2010 NAR 38:7432), which should be noted here. Indeed effects of NusG favoring forward translocation or disfavoring backtracking were first described in 2000 (Pasman and von Hippel, 2000 Biochemistry 39:5573) and were characterized in some detail by Turtola and Belogurov, 2016 *eLife* 5:e18096, which would be appropriate to cite.

5. Lines 123-15 I found the criteria for deciding when contributions of NusA or NusG to termination were significant confusing. The authors state that effects of Δ%T ≥ 25 were considered significant or to be negligible when 10 ≥ Δ%T ≥ -10. Because %T is a ratio that asymtoptically approaches 0 and 100, however, these criteria seem to ignore potentially large effects and magnify others based on the absolute %T. A change of 90% to 99% termination would apparently be considered negligible even though it is close to a 10-fold effect both on downstream gene expression and on the relative rates of termination vs. terminator readthrough. In contrast a change in from 25% to 50% termination is less than a 2-fold effect on downstream gene expression and about 3-fold on the relative rates of termination vs. terminator readthrough. Thus, in terms of mechanistically meaningful effects on NusA and NusG, the criteria chosen seem arbitrary and not based on an obvious rationale. Have the authors considered using criteria that more directly connect to mechanistic effects of NusA and NusG?

Reviewer #3:

New data in the past few years have led to profound new insights into the structure and function of the bacterial RNA polymerase complex. In particular, this information has begun to uncover fundamental mechanistic features of transcription elongation, and, by extension, transcription termination. NusG protein, when bound to the transcription elongation complex, is known to form transient interactions with additional factors, from Rho to r-protein S10. Recent data have also revealed interactions between NusG and NusA, although not through the NGN domain. NusA has been shown to promote termination at intrinsic terminator sites while NusG has been described as an anti-pausing factor that improves processivity of the elongation complex. *E. coli* is thought to exhibit close coupling between the transcription elongation complex and the leading translating ribosome, while an interesting recent study argued that transcription and translation are uncoupled in the Gram-positive model microorganism *Bacillus subtilis*. Therefore, transcription mechanisms may exhibit surprising differences between different phylogenetic groupings of bacteria. At the heart of this topic is NusG, which associates with a variety of factors in E coli, including NusA, NusB, S10 and Rho. Yet, B subtilis NusG appears to exhibit some differences, such as an apparent increase in sequence-specific NusG-DNA interactions, although the extent of these differences had not been examined. The current study addresses this issue and finds an entirely new mechanistic feature for B subtilis NusG proteins. High-throughput mapping of 3' termini can help identify sites of transcription termination at a global level. A prior study used this approach to discover the extent of NusA-mediated termination for *B. subtilis*. In this study, the authors repeat this approach with NusA- and NusG-depletion strains to find that "intrinsic termination" is somewhat of a misnomer, that the great majority of termination events appear to be strongly influenced by either NusA, NusG or both proteins. Moreover, their data point to some general trends, such as the length of the poly-uridine sequences and the overall size of terminator hairpin, which appear to be larger for truly intrinsic terminators and are weaker for the (more numerous) NusG/A-dependent terminators. This is a startling paper that fundamentally changes the perception of the role of NusG in *B. subtilis* and suggests that NusG and NusA work together to promote termination for bacteria where transcription and translation are uncoupled.

---

## [Author Response]

Essential revisions:1. While we appreciated the careful follow-up work, we felt that the major conclusion could be strengthened by a more in-depth analysis of the genome-wide data, assuming those data support the role of NusG-dependent pausing in termination. Reviewers 1 and 2 give specific suggestions in their reviews. Some of these relate to the way comparisons are made between datasets, and others address specific scientific questions. Of particular relevance are analyses that test whether NusG stimulates termination at sites with (i) weak terminal base-pairs, and (ii) gaps in the U-tract. Any other analyses of the genome-wide data that support the importance of NusG-simulated pausing in termination would be valuable to include. For example, is there any evidence that NusG-dependent pause sites identified by NET-seq are associated with sites of termination?

Before explaining our analysis of RNET-seq data, it should be noted that RNET-seq RNA isolation steps lead to a depletion of RNET-seq read coverage at intrinsic terminators. As such, we found that the intrinsic terminators identified by RNET-seq tended to have stronger RNA:DNA hybrids and stronger NusG-dependent pause signals compared to the total pool of intrinsic terminators, as these features stabilize RNAP at the POT. These findings complicated the direct comparison of RNET-seq data to Term-seq data.

This caveat is specified in lines 302-307 of the revised text.

(i) Weak terminal base-pairs, and (ii) gaps in the U-tract

The argument that a feature of NusG-dependent terminators with weak terminal base-pairs in the hairpin and distal U-rich tract interruptions was derived from the observation that the 3’ ends of all terminators tested in vitro were downstream of the 3’ ends identified in vivo. We suggested that this is due to post-release trimming by 3’ to 5’ exoribonucleases (added citation to PMID: 30381461). Regardless, in each case, after revising the U-rich tract and hairpin to account for this discrepancy, we noticed that each terminator tested in vitro contained several terminal A-U/G-U base-pairs and distal U-rich tract interruptions. Unfortunately, the discrepancy between the in vivo and in vitro 3' ends did not allow us to analyze whether this phenomenon is wide-spread at a transcriptome-wide level using only the Term-seq dataset presented in this manuscript.

RNET-seq reads are strictly obtained from nascent transcription events. Thus, the RNA 3’ ends identified by RNET-seq are unaffected from the effects of 3'-to-5' exonucleolytic trimming by ribonucleases. To determine how closely the RNET-seq and Term-seq derived 3’ ends agreed, we searched for the most abundant RNET-seq 3’ end (see revised Materials and methods) at, or 1-4 nt downstream of, the identified Term-seq 3’ end for each intrinsic terminator. For this task, we used the RNET-seq dataset in Yakhnin et al. 2020 collected from the WT strain of *B. subtilis*. Through this analysis, we identified 485 intrinsic terminators with an RNET-seq 3’ end at or 1-4 nt downstream of the 3’ end identified by Term-seq. Notably, for 82% of these intrinsic terminators, the RNET-seq 3’ end was 1-4 nt downstream of the Term-seq 3’ end. While this information does not directly connect exoribonuclease trimming to the 3' ends identified by Term-seq, it provides compelling evidence that the nascent 3’ ends of intrinsic terminators are consistently downstream of the steady-state 3’ ends of released transcripts.

The 485 intrinsic terminators identified by RNET-seq were sorted into those that terminated strongly and independently of NusG (%T ≥ 70 in the WT strain, Δ%T between 10 and -10 in the Δ*nusG* strain), and those with a Δ%T ≥ 25 in the Δ*nusG* strain (NusG-dependent). For the terminators in each of these categories, we revised the terminator hairpin stems based on the genomic coordinates of the RNET-seq derived 3’ ends and tabulated the number of terminators that contained ≥ 1 consecutive terminal A-U/G-U base pairs vs. those that had 0 terminal A-U/G-U base pairs. Conducting a Fisher’s exact test for count data on these groups revealed that terminators with a Δ%T ≥ 25 in the Δ*nusG* strain (NusG-dependent) were more likely to contain ≥ 1 consecutive terminal A-U/G-U base pairs compared to SI terminators. Next, Supplementary file 2 was revised to account for the RNET-seq derived 3’ ends and sequence logos were generated for the updated U-rich tracts of SI of NusG and NusG-dependent intrinsic terminators. Using the computational sequence logo comparison toolkit DiffLogo (PMID: 26577052), we visualized the per-nucleotide differences between the sequence logos generated for these 3 groups and found that an increase in NusG-dependence was correlated with a decrease in distal U-richness of the U-rich tract. This analysis provides transcriptome-wide evidence that NusG-dependent terminators commonly have ≥ 1 consecutive terminal A-U/G-U base pairs in the hairpin stem and distal U-rich tract interruptions.

The description of this analysis is in lines 277-296 of the revised manuscript. Supplementary file 4 and Figure 4—figure supplement 2 were added

Is there any evidence that NusG-dependent pause sites identified by NET-seq are associated with sites of termination?

A NusG-dependent pause site is defined as a transiently stabilized 3’ end that is absent or drastically reduced in abundance upon the loss of NusG. RNET-seq can be used to both identify all pause sites and quantify their NusG-dependence (Yakhnin et al. 2020). Thus, we calculated the normalized abundance of each of the previously mentioned 485 intrinsic terminator 3’ ends identified by RNET-seq in both the WT and Δ*nusG* strains (see revised Materials and methods). By calculating the log_2_ transformed WT: Δ*nusG* abundance ratio for each of these 3’ ends, and by sorting these values into those derived from SI of NusG and NusG-dependent terminators, we observed that NusG-dependent pausing could be found at both SI of NusG and NusG-dependent terminators. This finding is not surprising considering that a major component of the NusG-dependent pause motif are T residues on the nt-DNA strand at positions -6, -7, and -8, , which correspond to U residues at positions 2, 3, and 4 of the U-rich tract. It is likely that the only terminators that require this pause to terminate efficiently are suboptimal terminators with ≥ 1 consecutive terminal A-U/G-U base pairs in the hairpin stem, weaker hairpin stems, and/or distal U-rich tract interruptions.

The manuscript was updated to include this analysis on lines 297-302 and Figure 4—figure supplement 3 was added.

2. Termination sites in vitro are consistently downstream of those observed in vivo. While it is reasonable to hypothesize that this difference is due to trimming by exonucleases, there is no experimental evidence presented to support this. To test the hypothesis, we suggest mapping termination sites by 3' RACE in RNase mutant strains for one or two of the terminators characterized in the paper. B subtilis 3' exonucleases are defined, and mutant strains have been described (e.g., Oussenko et al., 2005 J Bacteriol 187:2758; Liu et al., 2014 Mol Microbiol 94:41).

We have RNET-seq information showing observable 3’ ends 1-4 nt downstream of the Term-seq identified 3’ ends for 82% of all intrinsic terminators identified by RNET-seq. The 3’ ends identified by RNET-seq do not always correspond precisely to the 3’ ends identified in vitro. Likely, the discrepancy between the 3’ ends identified in vitro and the 3’ ends identified via two in vivo methodologies are due to inherent differences between in vitro and in vivo conditions. For example, YhaM was found to trim several nt off of the 3’ ends of intrinsically terminated transcripts in the related firmicute *Streptococcus pyogenes,* which may explain the 1-4 nt difference between many nascent and released 3’ ends (Lécrivain et al. 2018).

The manuscript was updated to include this information on Lines 265-281 and Lines 431-437. The pertinent citation was added and Figure 4—figure supplement 2 was added.

We attempted to do the 3' RACE analysis as suggested, but we were unsuccessful in obtaining 3' RACE information using two different methods. Since this is a minor point of the manuscript, we feel that the new RNET-seq information is sufficient.

3. Add a figure showing the model described in the discussion (lines 332-84) for the proposed roles of NusG and NusA in intrinsic termination.

A figure showing our model is now included as figure 7.

4. Broaden the discussion of how the study relates to prior work on intrinsic termination, as detailed by Reviewer 2.

This was done and the manuscript has been edited to reflect these changes.

Reviewer #1:[…] Finding a widespread role for NusG in intrinsic termination is interesting, and the authors make a reasonably strong case that this effect is due to NusG-dependent pausing. Nonetheless, several of the arguments made would be strengthened by further analysis of the genome-wide datasets. The effects of losing NusA/G on gene expression are somewhat less interesting; these changes are not surprising for such important regulators. Moreover, the physiological significance of these changes is unclear. Do NusG levels fluctuate sufficiently for there to be changes in gene expression? I think it is unlikely that NusG levels would ever drop sufficiently to impact motility.There are a lot of comparisons of distributions in the manuscript. For the most part, the differences are clear, but a statistical analysis is warranted in the less obvious cases, e.g. Supplementary Figure 3.

Appropriate statistical analyses were conducted for all comparisons. For comparisons in which multiple pairwise analyses were conducted. The main text was modified to include information regarding statistical tests conducted and Supplementary file 3 was added to show the p-values derived from each pairwise comparison.

Figure 4 shows a few examples of NusG stimulating termination at sites with weak terminal base-pairs. This is an interesting result, but with only a few examples it is difficult to judge how widespread this is. The authors should analyze their genome-wide datasets to determine if this is a common feature of NusG-dependent terminators.

This was done as described above. It is an enriched feature.

Line 225. The authors argue that NusG stimulates termination at sites with gaps in the U-tract. This is an intriguing hypothesis, but should be tested using the genome-wide data.

This was done as described above. It is an enriched feature.

Reviewer #2:[…] Overall, this comprehensive analysis of terminators in Bacillus is highly informative, and the analysis of the NusA/NusG-dependence of these terminators provides important insights into transcription regulatory mechanisms. Nonetheless, the manuscript could be improved by addressing the following main points.1. The authors demonstrate a correlational but not causal relationship between NusG stimulation of pausing and its effect to enhance termination. They show that the pause exists without termination, but do not directly link pausing to termination. Have the authors identified mutations of NusG that still interact with RNAP but are defective for pause stimulation? Was a mutant NusG defective for the NT-strand interaction tested for the ability to promote termination? Unless more supportive data can be provided, the claim of causality should be tempered and the need for testing causality should be described.

This was an excellent suggestion. We previously demonstrated that a Y77H/N81S/T82V mutant NGN domain lost its ability to stimulate pausing because it no longer interacted with the ntDNA strand (Yakhnin et al. (2016) JBC 291:5299-5308). We purified this mutant NGN domain and found that it was unable to stimulate termination of the *yetJ* and *ktrD* terminators. The results of this experiment demonstrate a causal relationship between NusG-dependent pausing and NusG-dependent stimulation of intrinsic termination. The results of this experiment are described in the text in Lines 193-197 and 201-204 and are included in the revised Figure 2.

2. Section starting at line 162 and Figure 3. Related to point 1, the authors perform a detailed analysis of the ktrD terminator and focus on possible effects of NusG on pausing downstream from the site of termination mapped by term-seq (A+8 numbered from the last G-C base pair in the terminator hairpin). The authors conclude that NusG exerts its effect on intrinsic termination through its role in pausing (line 177), but do not highlight in the Results the fact that they do not observe pausing or termination at A+8 in vitro. If the effects of NusG on pausing all occur after the point of termination in vivo, then how could pauses at those locations possibly affect termination in vivo? The authors should address this question explicitly in the Results, since it seems to contradict their conclusion. The same point appears to apply to the yetJ and fur terminators (Figure 4). In the Discussion the authors suggest the discrepancy could be related to nuclease trimming in vivo of RNAs terminated at the sites seen in vitro. Readers will already be confused by the discrepancy in the Results, so it would better to bring up this idea, which is reasonable, in the Results. Have the authors considered testing it, however? Can nuclease mutants be used to see if pausing and termination really correlate in vivo and in vitro? Here again, unless a more definitive test is performed, the conclusions should highlight where uncertainty remains and be framed around the need for additional experiments.

The discrepancy between the in vivo and in vitro 3’ ends was further addressed as described above via RNET-seq and the manuscript has been edited to reflect this. Our attempts at 3' RACE were unsuccessful. We spent two months on this to no avail. Unfortunately, there is no way to use nuclease mutants to correlate pausing and termination in vivo.

3. The comparisons of different classes of terminators (stimulated by NusA, NusG, both, or neither; Figures 1, S3, S8) are reported using box plots or sequence logos without any accompanying statistical analysis for the significance of differences or similarities of the characteristics catalogued. Additionally, the box plots are ineffective in illustrating differences in distributions relative to the more information-rich violin plots or ridge plots that could be generated for the same datasets (e.g., using the R function geom_violin from the R package ggplot2 and the R function geom_density_ridges from the R package ggridges). Box plots work best when data are unimodal and roughly symmetric, which these data are not.Additionally, when making statements about comparisons of characteristics between large datasets, it is accepted practice to include an estimate of the statistical significance of any differences described (i.e., a p value, although more sophisticated estimates of statistical significance are also possible). The authors should consider alternative presentations distributions (e.g., of %T) and should include estimates of statistical significance for any comparison from which they draw conclusions. For example, the change in %T from WT could easily be tested for statistical significance using a Wilcoxon signed-rank test (a non-parametric paired difference test that avoids distributional assumptions). The distribution of predicted terminator hairpin stem lengths could be evaluated using a permutation test of shuffled classes.

These were excellent suggestions. While we decided to keep the box plots for all circumstances, we also overlayed violin plots onto all box plots to show the shape of the data in Figure 1, Figure 1—figure supplement 4, Figure 4—figure supplement 3. We also conducted appropriate statistical tests whenever making comparisons as described above. We also used DiffLogo to compare sequence logos as described above.

4. The manuscript could be improved by placing the results in the broader context of what is already known and reported in the literature about intrinsic termination, NusA, and NusG. Currently, the authors' description of prior knowledge is more solipsistic than comprehensive. This narrow perspective limits in the impact of the work and at some points is misleading about what is already known or in debate. Examples of omissions are listed here. Correcting them comprehensively would greatly improve the manuscript.(a) The authors do not cite a major prior study of genome-scale intrinsic termination in *B. subtilis* (Lalanne et al., 2018 Cell 173:749). They should compare their results to those of Lalanne et al. at multiple places in the manuscript. In validating their term-seq data (lines 109-115), the authors should include a comparison of their findings to those of Lalanne et al.

We erred in not citing this paper, especially considering that we have discussed the Rend-seq method at length. This important paper is cited in the revised manuscript. We also compared our new terminator data with that from the Lalanne paper. We identified 1123 intrinsic terminators shared between this work and Lalanne et al. 2018. This was included in the main text in Lines 124-126.

In the analysis of misregulation by NusA/NusG depletion/deletion, it seems obvious to ask whether NusA, NusG, or both contribute significantly to the 167 intergenic, 'tuned' intrinsic terminators reported by Lalanne et al. to singly or in combination set the expression of 276 genes in *B. subtilis*. To what extent do contributions of NusA and NusG to termination at these 167 sites explain the misregulation of gene expression reported in the current manuscript?

We first determined the % of all intrinsic terminators that were determined to be SI, req A, req G, req A and G, and req A or G. We then determined the % of all ‘tuned’ intrinsic terminators that were determined to be SI, req A, req G, req A and G, and req A or G. By conducting pairwise Fisher’s exact test for count data, we found no difference between these proportions when comparing all intrinsic terminators to ‘tuned’ intrinsic terminators specifically. Thus, we have no reason to suspect that NusA and/or NusG preferentially contributes to the efficiency of this class of intrinsic terminators.

(b) (line 96) As best I can tell, term-seq was developed independently by the Babitzke group and by the Sorek group in Israel (Dar et al., 2016 Science 352:aad9822). However, Dar et al. are not cited when describing the method. It would be appropriate to cite both the authors' own work and Dar et al. at this point in the manuscript.

The reviewer is correct-the two labs developed the method independently. The Dar et al. paper is now cited as suggested.

(c) At multiple places in the manuscript (abstract, line72, line 314), the authors suggest that current or general dogma is that intrinsic terminators are not affected by transcription factors (or perhaps they mean not significantly affected by transcription factors). These statements employ a common trope in the scientific literature to frame and oversimplify prior knowledge so as to increase the perceived novelty of findings, and often constitute a strawman in scientific rhetoric. In this case, stating that intrinsic terminators not being significantly affected by transcription factors is a general dogma ignores long-standing knowledge that NusA stimulates intrinsic termination. Indeed NusA was discovered as an intrinsic termination factor and has long been described as such. See, for example, Greenblatt et al., 1980 PNAS 77:1991 "L factor that is required for β-galactosidase synthesis is the nusA gene product involved in transcription termination"; Schmidt and Chamberlin, 1987 JMB 195:809 "nusA protein of *Escherichia coli* is an efficient transcription termination factor for certain terminator sites"; Kingston and Chamberlin, 1981 Cell 27:523, which states in the abstract "termination at [rrnB] tL is dependent on the nusA protein"; Farnham et al., 1982 Cell 29:945 "Effects of NusA protein on transcription termination in the tryptophan operon of *Escherichia coli*" or, more recently, Gusarov and Nudler, 2001 Cell 107:437 "Control of intrinsic transcription termination by N and NusA: the basic mechanisms." Thus, there is no dogma that intrinsic terminators are not affected by transcription factors. There may be confusion about the definition of intrinsic termination, which is by no means exclusive to this manuscript. Intrinsic terminators are terminators that do not require transcription factors to exhibit at least some level of termination; they are not defined as terminators that are not affected by transcription factors. It is accurate to say the characteristics of intrinsic terminators that are affected by NusA (or in this case NusA and NusG) are not well characterized. The authors' results make a major contribution to such characterization, but they should be presented in that context and rather than the tired trope of overthrowing a dogma that doesn't really exist.

Point taken. We removed any suggestion of overthrowing dogma. We are fully aware of the in vitro literature of NusA-stimulated termination and cite some of this work. Our previously published work remains as the only demonstration that NusA-dependent/NusA-stimulated termination occurs in vivo. Prior to our current work, the only suggestion that NusG affected intrinsic termination that we are aware of is one publication from the Landick lab using Mycobacterial RNAP in vitro. That work was cited in the Discussion of the original version of this manuscript and is now cited in the Introduction as well (Lines 79-80).

The Abstract was modified to emphasize that the information pertains specifically to *B. subtilis.* We assumed that this was implied considering that *Bacillus subtilis* is specifically mentioned in the preceding sentence.

These *E. coli* references were added to the description pertaining to line 72. (Greenblatt et al., 1981; Schmidt et al., 1987; Bermúdez-Cruz et al., 1999).

(d) At multiple places in the manuscript (line 15, line 72, line 302), the authors suggest that NusG was not previously known to affect transcriptional termination. This statement also is misleading. Mycobacterial NusG is known to stimulate intrinsic termination by Mycobacterial RNA polymerase and to stimulate intrinsic termination additively with NusA at some terminators (Czyz et al., 2016 mBio 5: e00931). The authors should include this precedent when introducing NusG effects on termination.

Line 15 of the Abstract was modified to emphasize that the information pertains specifically to *B. subtilis.* We previously cited the 2014 Czyz et al. work at the end of the Discussion, but now mention these findings in the Introduction, and the sentence pertaining to line 302 was modified.

(e) (line 334) The authors note that "NusG shifts RNAP to the post-translocation register". This was previously reported (Sevostyanova and Artsimovitch, 2010 NAR 38:7432), which should be noted here. Indeed effects of NusG favoring forward translocation or disfavoring backtracking were first described in 2000 (Pasman and von Hippel, 2000 Biochemistry 39:5573) and were characterized in some detail by Turtola and Belogurov, 2016 eLife 5:e18096, which would be appropriate to cite.

Sevostyanova and Artsimovitch, 2010 NAR 38:7432 is cited in the revised manuscript. Backtracking is not mentioned in our manuscript and so we did not cite the other two references.

5. Lines 123-15 I found the criteria for deciding when contributions of NusA or NusG to termination were significant confusing. The authors state that effects of Δ%T ≥ 25 were considered significant or to be negligible when 10 ≥ Δ%T ≥ -10. Because %T is a ratio that asymtoptically approaches 0 and 100, however, these criteria seem to ignore potentially large effects and magnify others based on the absolute %T. A change of 90% to 99% termination would apparently be considered negligible even though it is close to a 10-fold effect both on downstream gene expression and on the relative rates of termination vs. terminator readthrough. In contrast a change in from 25% to 50% termination is less than a 2-fold effect on downstream gene expression and about 3-fold on the relative rates of termination vs. terminator readthrough. Thus, in terms of mechanistically meaningful effects on NusA and NusG, the criteria chosen seem arbitrary and not based on an obvious rationale. Have the authors considered using criteria that more directly connect to mechanistic effects of NusA and NusG?

In our 2016 Nature Microbiology paper on NusA-dependent termination we defined a ∆%T ≥ 25 as being NusA-dependent, and ∆%T between 10 and 25 as NusA-stimulated. We recognize that these cut-offs are arbitrary, but it allowed us to categorize the data so that we could discuss the results in a more meaningful way. We debated at great length about whether we should use fold change vs. ∆%T for the very reason mentioned by the reviewer (termination vs. readthrough). The various classes of terminator defects and the relative dependence on NusA fit the ∆%T scheme. Moreover, the work focused on the mechanism of termination (∆%T) rather than effects on downstream gene expression (termination vs. readthrough). We feel that this nomenclature is also the best way to describe the effects of NusG so we continued with this nomenclature in the current manuscript.